# Brahma is essential for *Drosophila* intestinal stem cell proliferation and regulated by Hippo signaling

Yunyun Jin[1], Jinjin Xu[1], Meng-Xin Yin[1], Yi Lu[1], Lianxin Hu[1], Peixue Li[1], Peng Zhang[2], Zengqiang Yuan[2], Margaret S Ho[3], Hongbin Ji[1], Yun Zhao[1]*, Lei Zhang[1]*

[1]State Key Laboratory of Cell Biology, Institute of Biochemistry and Cell Biology, Shanghai Institutes for Biological Sciences, Chinese Academy of Sciences, Shanghai, China; [2]State Key Laboratory of Brain and Cognitive Science, Institute of Biophysics, Chinese Academy of Sciences, Beijing, China; [3]Department of Anatomy and Neurobiology, Tongji University School of Medicine, Shanghai, China

**Abstract** Chromatin remodeling processes are among the most important regulatory mechanisms in controlling cell proliferation and regeneration. *Drosophila* intestinal stem cells (ISCs) exhibit self-renewal potentials, maintain tissue homeostasis, and serve as an excellent model for studying cell growth and regeneration. In this study, we show that Brahma (Brm) chromatin-remodeling complex is required for ISC proliferation and damage-induced midgut regeneration in a lineage-specific manner. ISCs and enteroblasts exhibit high levels of Brm proteins; and without Brm, ISC proliferation and differentiation are impaired. Importantly, the Brm complex participates in ISC proliferation induced by the Scalloped–Yorkie transcriptional complex and that the Hippo (Hpo) signaling pathway directly restricted ISC proliferation by regulating Brm protein levels by inducing caspase-dependent cleavage of Brm. The cleavage resistant form of Brm protein promoted ISC proliferation. Our findings highlighted the importance of Hpo signaling in regulating epigenetic components such as Brm to control downstream transcription and hence ISC proliferation.

*For correspondence: rayzhang@ sibcb.ac.cn (LZ); yunzhao@sibcb. ac.cn (YZ)

**Competing interests:** The authors declare that no competing interests exist.

**Reviewing editor**: K Vijay Raghavan, National Centre for Biological Sciences, Tata Institute of Fundamental Research, India

## Introduction

Central to the animal development is how chromatin assembly and regulation orchestrate cell-fate determination. Four epigenetic factors, DNA methylation, histone modifications, ATP-dependent chromatin remodeling, and the recently discovered non-coding RNAs play major roles in epigenetic regulation at the chromatin level. The SWI/SNF family is one of the most-studied families of ATP-dependent chromatin remodeling complexes, which regulate gene expression by destabilizing nucleosome structures to alter the DNA accessibility for transcription factors (*Cairns, 2007*; *Hargreaves and Crabtree, 2011*). Studies have implicated diverse roles for the mammalian SWI/SNF complexes in embryonic stem cell proliferation and differentiation. SWI/SNF complexes also function in neural, heart, and muscle development (*Bultman et al., 2000*; *Lickert et al., 2004*; *Ho et al., 2009*; *Ho and Crabtree, 2010*; *Zhan et al., 2011*). In *Drosophila*, there are two SWI/SNF complexes, the Brahma (Brm)-associated proteins (BAP) complex and the polybromo-containing BAP (PBAP) complex. The BAP complex has a signature subunit Osa, while PBAP complex is defined by BAP170, Polybromo, and Syap (*Elfring et al., 1998*; *Chalkley et al., 2008*). Brm is a unique DNA-stimulated ATPase and common subunit for both BAP and PBAP complexes. Progress has been made in understanding the function of the Brm complex during *Drosophila* development (*Treisman et al., 1997*; *Collins and Treisman, 2000*; *Janody et al., 2004*; *Moshkin et al., 2007*; *Carrera et al., 2008*; *Terriente-Felix and de Celis, 2009*; *Neumuller et al., 2011*), yet little is known about Brm complex functions in maintaining stem cell pluripotency of the epithelial tissues.

**eLife digest** Most tissues can generate new cells to repair damage or replace worn-out cells. The new cells are often generated from stem cells—cells that can either reproduce themselves or mature into other types of cells. In the fruit-fly *Drosophila*, for example, intestinal stem cells in the midgut are capable of producing more stem cells or they can differentiate to produce immature cells called enteroblasts that go on to become either enterocytes (the cells that line the gut) or enteroendocrine cells (which secrete hormones).

Researchers have identified a number of signalling pathways that are involved in the proliferation and differentiation of intestinal stem cells in the midgut of fruit flies. These include the Hippo pathway, which is important for regulating both cell proliferation and programmed cell death (apoptosis). Activation of the Hippo protein triggers a cascade of signals that culminate in the regulation of many of the genes involved in cell proliferation, division and apoptosis.

Another process that is important for controlling the proliferation and differentiation of cells is chromatin remodelling. Chromatin is the 'packaging' that keeps DNA tightly wound within the cell nucleus, and remodelling refers to the structural changes that allow proteins called transcription factors to reach the genes and transcribe them into messenger RNA (which then leaves the nucleus to generate the protein).

Now, Jin et al. have explored how the Hippo pathway and chromatin remodelling work together to regulate of stem cells. Using a technique called RNA interference to block the expression of various genes in intestinal stem cells and enteroblasts, Jin et al. found that a protein called Brahma—which is an essential part of a chromatin-remodelling complex—must be present for the stem cells to multiply normally.

Jin et al. also showed how the Hippo signalling pathway interacts with chromatin remodelling. Activation of the Hippo pathway inhibits gene expression by preventing two other proteins, Yorkie and Scalloped, from forming a complex in the nucleus. The new work shows that Brahma interacts physically with the Yorkie and Scalloped proteins to regulate the proliferation of the intestinal stem cells. It also shows that the Hippo protein regulates the activity of the Brahma protein by inducing a process called caspase-dependent cleavage. Because many of the proteins involved in these pathways are evolutionarily conserved and expressed in a variety of tissues, these findings may have implications for stem cell function and tissue repair in many species.

The simplicity of the structure and the multipotency of *Drosophila* posterior midgut make it an excellent model to study adult epithelial tissue homeostasis and regeneration (*Micchelli and Perrimon, 2006*; *Ohlstein and Spradling, 2006*). The midgut is composed of four cell types: enterocytes (ECs), enteroendocrine (ee) cells, intestinal stem cells (ISCs), and enteroblasts (EBs). The mature ECs are large polyploid cells of absorptive function and frame the midgut lining; ee and ISCs are the two types of diploid cells in the midgut that are less abundant. ISCs evenly locate at basal position underneath the ECs with a wedge-like morphology (*Ohlstein and Spradling, 2006*, *2007*) and are the only known cell type in the posterior midgut that proliferates. On cell division, ISCs undergo self-renewal or proliferation to become EBs, quiescent progenitor cells that ultimately differentiate to ECs or ee cells with the ratio 9:1 under the control of Delta (Dl) and Notch (*Micchelli and Perrimon, 2006*; *Ohlstein and Spradling, 2006*). Since the active Dl expression is retained in self-renewed ISCs and is lost in the newly generated EBs, antibody against the active Dl is used as the specific and the only known marker for *Drosophila* ISCs (*Ohlstein and Spradling, 2007*). It has been demonstrated that the proliferation and differentiation of ISCs are tightly controlled by Notch, Janus kinase/signal transducer and activator of transcription (JAK/STAT), epidermal growth factor receptor/mitogen-activated protein kinase (EGFR), Hippo (Hpo), and Wingless signaling pathways (*Jiang and Edgar, 2011*).

The evolutionarily conserved Hpo pathway controls organ size by regulating cell proliferation and apoptosis (*Pan, 2010*; *Halder and Johnson, 2011*; *Yin and Zhang, 2011*; *Irvine, 2012*). Hpo is a serine/threonine Ste20-like kinase (*Harvey et al., 2003*; *Jia et al., 2003*; *Pantalacci et al., 2003*; *Udan et al., 2003*; *Wu et al., 2003*) that directly phosphorylates and activates downstream nuclear Dbf2-related (NDR) family protein kinase Warts (Wts). Wts activation mediated by Hpo requires scaffold proteins Salvador (Sav) (*Kango-Singh et al., 2002*; *Tapon et al., 2002*) and mob as tumor suppressor

(Mats) (*Lai et al., 2005*). Together, these proteins inhibit Yorkie (Yki) nuclear translocation. In the absence of Wts-mediated suppression, Yki forms a complex with transcription factor(s) such as Scalloped (Sd) (*Goulev et al., 2008*; *Wu et al., 2008*; *Zhang et al., 2008*) in the nucleus to regulate the expression of a plethora of genes involved in cell proliferation, cell cycle progression, and apoptosis (*Halder and Johnson, 2011*; *Yin and Zhang, 2011*; *Irvine, 2012*). In addition, the Hpo pathway maintains tissue homeostasis by regulating the balance between *diap1* expression and basal levels of activated caspases via the control of Dronc (*Drosophila* Nedd-2-like caspase orthologous to human Caspase 9) (*Verghese et al., 2012*).

We present evidence that Brm is required for ISC proliferation in both normal and regenerating midguts, and it is required in ISCs for EC differentiation in normal midguts. In addition, we show that the Brm complex is physically associated with the Sd–Yki transcriptional complex in the nucleus and functions downstream of the Hpo pathway to regulate ISC proliferation. We also demonstrate that Brm is regulated by the Hpo pathway at the protein level through Hpo kinase-induced, caspase-dependent, cleavage of Brm at its D718 site. Altogether, as exemplified in the *Drosophila* ISCs, our study unravels a novel mechanism of the chromatin remodeling Brm complex in maintaining adult stem cell pluripotency of epithelial tissues.

## Results

### Brm is required for ISC proliferation in midguts

To gain insights on homeostasis and proliferation of *Drosophila* midguts, a small-scale screen searching for candidates that genetically alters the midgut regeneration and homeostasis was carried out. During the screen, *escargot-Gal4* (*esg-Gal4*) was used to drive RNAi expressions of different genes in ISCs and EBs in the presence of a temperature-sensitive Gal4 repressor, *tubGal80* (henceforth *esg80$^{ts}$*). Adult *esg80$^{ts}$* flies grown at the permissive temperature do not express GFP or RNAi in ISCs and EBs. Once shifted to the non-permissive temperature, RNAi expressions in ISCs and EBs are induced and simultaneously marked by *esg-Gal4*-driven GFP signals (*Micchelli and Perrimon, 2006*). Interestingly, among the RNAi lines, VDRC (37720) and Bloomington (31712) abolished the expression of Brm, the energy providing subunit in *Drosophila* Brm complex (*Neumuller et al., 2011*; *Waldholm et al., 2011*). On Brm RNAi expression, the number of GFP positive (GFP$^+$) cells in the adult posterior midgut decreased. Concomitantly, the number of phospho-histone3 positive (PH3$^+$) cells also reduced, suggesting that ISC proliferation is affected (compare *Figure 1B,B'* with *Figure 1A,A'*, also *Figure 1E*). Immunostaining using an antibody against Brm 505–775 aa (*Elfring et al., 1998*) confirmed that endogenous Brm protein can be efficiently knocked down in the cells of both wing imaginal discs and midguts that express Brm RNAi transgenes (compare *Figure 1—figure supplement 1B–B''*,*C–C'* with *Figure 1—figure supplement 1A*; and compare *Figure 1—figure supplement 1E–E'''* with *Figure 1—figure supplement 1D–D''*). In addition, GFP$^+$ cells exhibited a spherical shape in the absence of Brm compared with the angular shaped control cells (compare *Figure 1B,B'* with *Figure 1A,A'*), suggesting that the attachment of GFP$^+$ cells to surrounding cells is disrupted. We further tested whether knockdown of Brm in ISCs/EBs affects the division of ISCs. On Brm RNAi expression, EBs in the intestinal epithelium labeled with the expression of the Suppressor of Hairless reporter (Su(H)-LacZ, a specific marker for EBs) (*Micchelli and Perrimon, 2006*) were detected (compare *Figure 1—figure supplement 1G–G''* with *Figure 1—figure supplement 1F–F''*). This piece of evidence suggests that EBs are still formed even when Brm expression is inhibited and ISC proliferation is blocked. Expression of Brm$^{K804R}$, a dominant negative form of Brm defective for ATP hydrolysis activity without affecting the complex assembly (*Elfring et al., 1998*), results in similar effects compared to Brm RNAi (*Figure 1C,C'* and *Figure 1E*). Of note, we observed a mild increase in the ISC/EB numbers on Brm overexpression, and the PH3$^+$ cell number was slightly increased (compare *Figure 1D,D'* with *Figure 1A,A'*,*E*).

Interestingly, ISC/EB reduction induced by the loss of Brm might be due to an alternation in the rate of proliferation and differentiation. We hypothesized that the loss of Brm might result in an inhibition of ISC proliferation, precocious ISC differentiation, or a blockage of ISC differentiation. To test these possible mechanisms, the Mosaic analysis with a repressible cell marker (MARCM) approach (*Lee and Luo, 2001*) was used to generate *brm* null allele (*brm²*) clones, and its impact on midgut proliferation was analyzed (*Micchelli and Perrimon, 2006*; *Ohlstein and Spradling, 2006*). The wild-type MARCM stem cell clones divided indefinitely, their sizes increased linearly, and contained several or all midgut cell types (*Micchelli and Perrimon, 2006*; *Ohlstein and Spradling, 2006*). If Brm is important for ISC proliferation, the *brm²* clones will be retained in a limited size; if Brm is necessary for EC differentiation,

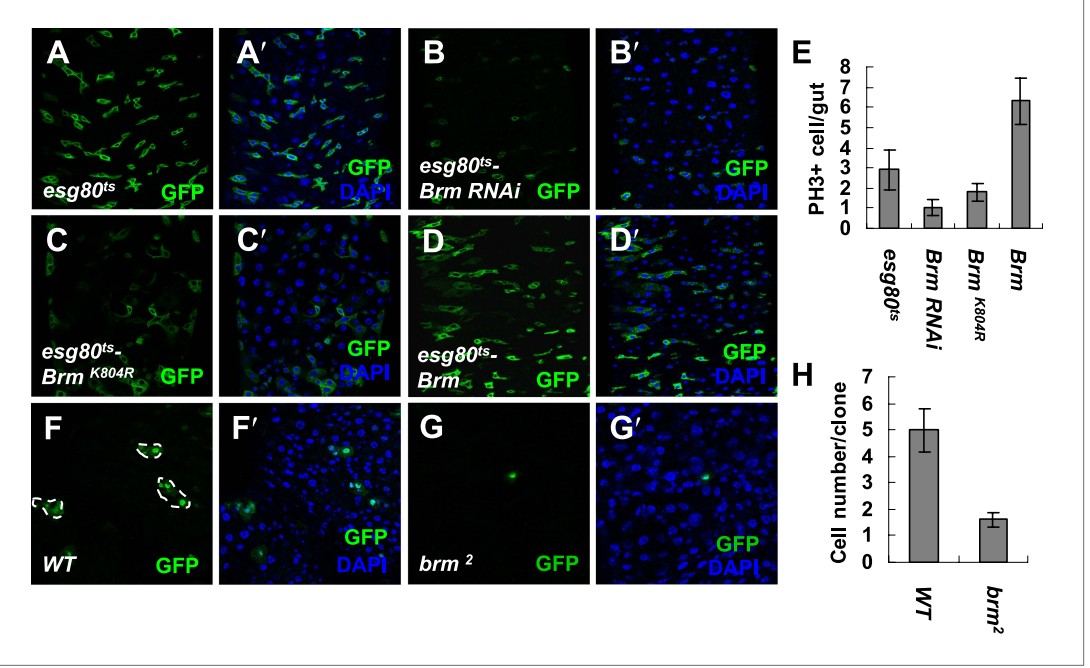

**Figure 1**. Brm is required for ISC proliferation in midguts. (**A**–**D'**) Adult fly midguts expressing *esg80ts-Gal4/ UAS-GFP* (*esg80ts*) (**A** and **A'**), *Brm RNAi* (*esg80ts-Brm RNAi*) (**B** and **B'**), *esg80ts-Gal4/UAS-GFP-BrmK804R* (*esg80ts-BrmK804R*) (**C** and **C'**) or *esg80ts-Gal4/UAS-GFP-Brm* (*esg80ts-Brm*) (**D** and **D'**) were immunostained with DAPI (blue). ISCs and EBs were marked by *esgGal4-driven* GFP expression. (**E**) Quantification of PH3+ cells of adult midguts of the indicated genotypes. The results represent the mean ± SEM, n = 10 for each genotype. (**F**–**G'**) Adult midguts containing nuclear localized GFP-labeled control MARCM clones (**F** and **F'**) or *brm* null allele *brm2* clones (**G** and **G'**) were immunostained for DAPI (blue). Guts were dissected from the adult flies 72 hr after clone induction. (**H**) Quantification of the cell numbers of the control or mutant clones of the indicated genotypes. The results represent the mean ± SEM, n = 10 for each genotype. See also *Figure 1—figure supplements 1 and 2*.

The following figure supplements are available for figure 1:

**Figure supplement 1**. Brm is required for ISC proliferation.

**Figure supplement 2**. Brm complex is required for ISC proliferation.

the *brm2* clones should mostly contain the small nuclear ISCs/EBs. Compared with the control clones that contain an average of five cells including both large nuclear cells and small nuclear cells within each clone, 3-day *brm2* clones contain only one or two cells, which are all small nuclear cells (*Figure 1H*, and compare *Figure 1G,G'* with *Figure 1F,F'*). In addition, 8-day *brm2* clones contain only one or two cells (*Figure 1—figure supplement 2A,A',B,B',C*). These results suggest that both proliferation of these clones and the EC differentiation are affected, suggesting that Brm is indispensable for ISC proliferation and EC differentiation in midguts.

We further tested the function of other subunits of the Brm complex in ISC proliferation. We found that the knockdown of other components in the Brm complex, including Bap60, Mor, and Osa by RNAi respectively under the control of *esg80ts* inhibited ISC proliferation to different extents and the GFP signal intensities were reduced simultaneously (compare *Figure 1—figure supplement 2F,F',H,H',J,J'* with *Figure 1—figure supplement 2D,D'*). Similar to Brm overexpression, overexpression of other Brm complex components induced only a mild enhancement on midgut ISC proliferation (compare *Figure 1—figure supplement 2E,E',G,G',I,I'* with *Figure 1—figure supplement 2D,D'*). In toto, these findings indicate that the maintenance of ISCs and EBs requires the presence of Brm complex.

## Brm is required for EC differentiation in normal midguts

Our results indicated that *brm2* clones only contained small nuclear cells (*Figure 1G,G'*), suggesting that Brm plays a role during ISC differentiation in addition to ISC proliferation. We first analyzed the

expression pattern of Brm during ISC cell maturation using *Myo1AGal4-GFP* (*Morgan et al., 1994*). *Myo1AGal4* is an enhancer trap in the gut-specific brush border *myosin 1A* gene that combined *tubGal80^ts* with the *Myo1AGal4* driver and *UAS-GFP* (together referred to as *Myo1A-GFP*). Interestingly, Brm antibody staining detected a high level of endogenous Brm proteins in ISCs/EBs (GFP⁻ cells in *Figure 2A–A'''* and GFP⁺ cells in *Figure 2B–B'''*), and some ee cells (co-labeled by *prospero*, a conserved homodomain transcription factor), whereas a relatively low level of Brm protein was detected in ECs (GFP⁺ cells in *Figure 2A–A'''*).

On the basis of these findings, we examined the role of Brm in ISC differentiation by overexpression or knockdown of Brm in the ISCs using a lineage induction system, *esg^ts F/O*. In this lineage tracing system, progenitor cells and their newborn progenies express *Gal4* and UAS-linked Gal4 targets, including the *UAS-GFP* marker (*Jiang et al., 2009*). PDM-1, a marker for fully differentiated ECs, is used to identify ECs (*Xu et al., 2011*). Overexpressing Brm for 2 days generated new EC-like GFP⁺ cells with large nuclei (*Figure 2D,D'*), whereas the wild type control group and the Brm RNAi group only contained GFP⁺ cells with small nuclei (*Figure 2C,C',E,E',I*). It is implicated that high levels of Brm lead to precocious differentiation of ISCs. After 5-day or even 13-day induction, large mature ECs were formed in wild-type midguts, while Brm RNAi suppressed ISC proliferation and EC differentiation in experimental midguts (compare *Figure 2H,H'* with *Figure 2F,F',I*, and *Figure 2—figure supplement 1A–C'*), suggesting that Brm is essential for ISCs and EBs to differentiate into ECs. In summary, the knockdown of Brm by RNAi blocks ISC proliferation and EC differentiation.

## Brm is required for midgut regeneration

Interestingly, in addition to its role in ISC proliferation under normal physiological context, Brm is also required for damage-induced midgut regeneration. Previous studies have reported that the feeding of dextran sulphate sodium (DSS) causes midgut cell proliferation via the disruption of basement membrane organization and increases in the intestinal stem cell division without affecting the final EB differentiation (*Amcheslavsky et al., 2009*). It is plausible to think that Brm also exerts an effect on DSS-induced midgut cell proliferation, as it is required for midgut cell proliferation. Indeed, when Brm RNAi was expressed, DSS-induced increase of GFP⁺ cells was blocked (compare *Figure 3C,C',D,D'* with *Figure 3A,A',B,B'*), suggesting that Brm is required in ISCs for DSS-induced proliferation. Of note, we did not observe dramatic change in ISC proliferation when overexpressing Brm in these GFP⁺ cells with or without DSS treatment (compare *Figure 3E,E',F,F'* with *Figure 3A,A',B,B'*).

## Brm interacts with the Hpo pathway effector Yki–Sd transcriptional complex

Previous studies implicated that the Hpo pathway effector Yki functions as a driver of proliferation in both ECs and ISCs and damage-induced ISC proliferation via both cell-autonomous and non-cell-autonomous mechanisms (*Karpowicz et al., 2010*; *Ren et al., 2010*; *Shaw et al., 2010*). Considering that Brm is involved in DSS-induced ISC proliferation (*Figure 3*), we tested whether there is a functional link between Brm and Yki–Sd transcriptional complex. To this end, mass spectrum (MS) analysis was first performed. Co-immunoprecipitation (Co-IP) experiments were performed in S2 cells to pull down the endogenous Yki or Sd protein using antibodies, and the pull-down products were sent for MS analysis. Several Brm complex components were found in the MS results, including Brm, Osa, Bap60, Bap55, and Mor (*Table 1*, see *Table 1* for MS details). Consistent with the results of Yki MS analysis (*Table 1*), we found that Yki and Brm coimmunoprecipitated with each other when Myc-tagged Yki (Myc–Yki) and V5-tagged Brm (Brm–V5) were coexpressed in S2 cells (*Figure 4A*). We also verified the interaction between Brm and Sd using Co-IP in S2 cells. Results showed that overexpressed HA-tagged Sd (HA–Sd) interacted with the endogenous Brm (*Figure 4B*). Sd also coimmunoprecipitated with Mor and Osa but not Bap60 when they were coexpressed in S2 cells (*Figure 4—figure supplement 1A*). In addition, we checked the cellular localization of these proteins in S2 cells. The majority of overexpressed Brm and overexpressed Sd were located in the nucleus (*Figure 4—figure supplement 1B–D'''*), whereas cytoplasmic–nuclear localization of Yki was not affected by Brm coexpression (data not shown), implicating that Brm complex does not promote the nuclear localization of Yki to influence the transcriptional activity of Yki–Sd complex.

## brm interacts with sd genetically

The genetic interaction between *brm* mutant (*brm²*) and *sd* hypomorphic allele (*sd¹*) in adult fly wings was examined. Strong mutations in *sd* cause lethality, while hypomorphic mutant *sd¹* flies are viable

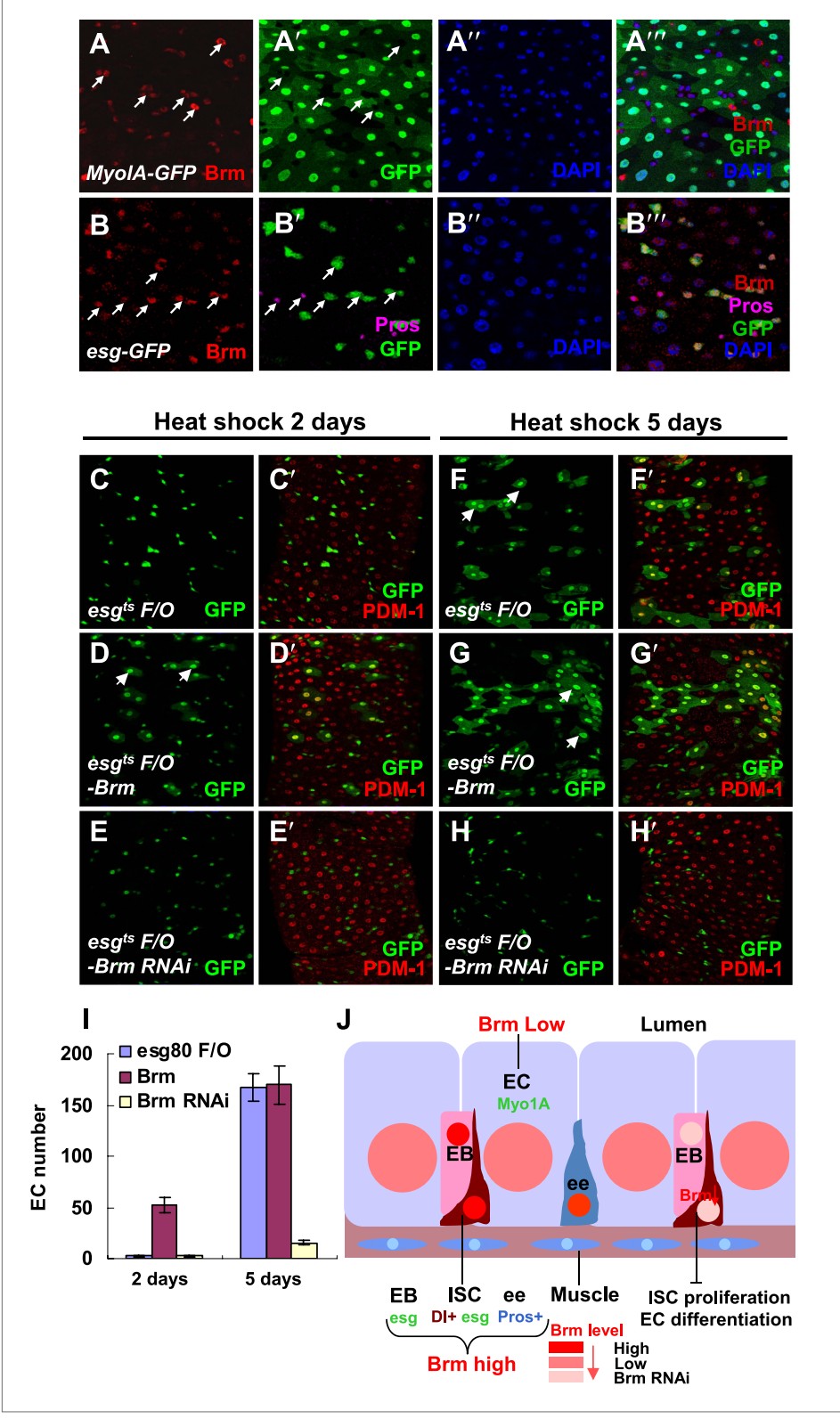

**Figure 2**. Brm is required for EC differentiation. (**A**–**B′′′**) Adult guts of wild-type *Myo1A-Gal4/UAS-GFP;tubGal80^{ts}* (**A**–**A′′′**) and *esgGal4/UAS-GFP* (**B**–**B′′′**) were immunostained with Brm antibody (indicated with arrows) to show the endogenous Brm protein level in the different cell types. (**C**–**H′**) Adult female midguts differentiation measured via the *esg^{ts} F/O* system. Transgenes were induced for 2 days (**C**–**E′**) or 5 days (**F**–**H′**). *esg^{ts} F/O-Brm* (**D**, **D′** and **G**, **G′**)

*Figure 2. Continued on next page*

*Figure 2. Continued*

promoted the formation of ECs, while *esg^{ts} F/O-Brm RNAi* (**E**, **E'** and **H**, **H'**) blocked the EC differentiation. ECs are marked by PDM-1 (red) and arrows. (**I**) Female posterior midguts were scored for GFP^+ and PDM-1^+ EC cells in the same region near the Malpighian tubules. The results represent the mean ± SEM, n = 10 for each genotype. (**J**) A schematic diagram of the regulation of Brm activity in intestinal homeostasis. ISCs divide asymmetrically to an EB and an ISC. EBs then differentiate into ECs or ee cells. Cell-type-specific markers are indicated. In normal state (left side), Brm is expressed at a high level in nuclei of ISCs, EBs, and some ee cells, and at a low level in nuclei of ECs. The different Brm protein levels in nuclei are marked by red (ISCs, EBs, and ee cells) or pink (ECs). Decrease of Brm protein level in ISCs reduces the ISC proliferative ability and inhibits EC differentiation (right). See also *Figure 2—figure supplement 1*.

The following figure supplements are available for figure 2:

**Figure supplement 1**. Brm is required for EC differentiation.

---

with a scalloped wing phenotype (compare *Figure 4E* with *Figure 4C*). Single-mutant *brm^2* fly wings are normal (*Figure 4D*). Interestingly, the double-mutant combination of *sd^1* and *brm^2* shows a strong enhancement of the *sd^1* phenotype (compare *Figure 4F* with *Figure 4E*). A similar enhanced phenotype was also found in the flies with *sd^1* and *osa^2*, a hypomorphic allele of a Brm complex subunit (compare *Figure 4H* with *Figure 4E,G*). These observations indicate that *brm* and *osa* genetically interact with *sd* and contribute to the wing vein alternation phenotype. Together with the biochemical results, these results suggest that Brm complex plays a crucial role in Yki–Sd mediated function.

## Brm functions downstream of Yki–Sd to maintain ISC proliferative ability

To further test whether Yki-mediated ISC proliferation depends on Brm, we examined the requirement of Brm activity during Yki–Sd induced ISC proliferation. Overexpression of either Yki or SdGA, an active form of Sd (*Zhang et al., 2008*), under the control of *esg80^{ts}* resulted in an increase in GFP^+ and PH3^+

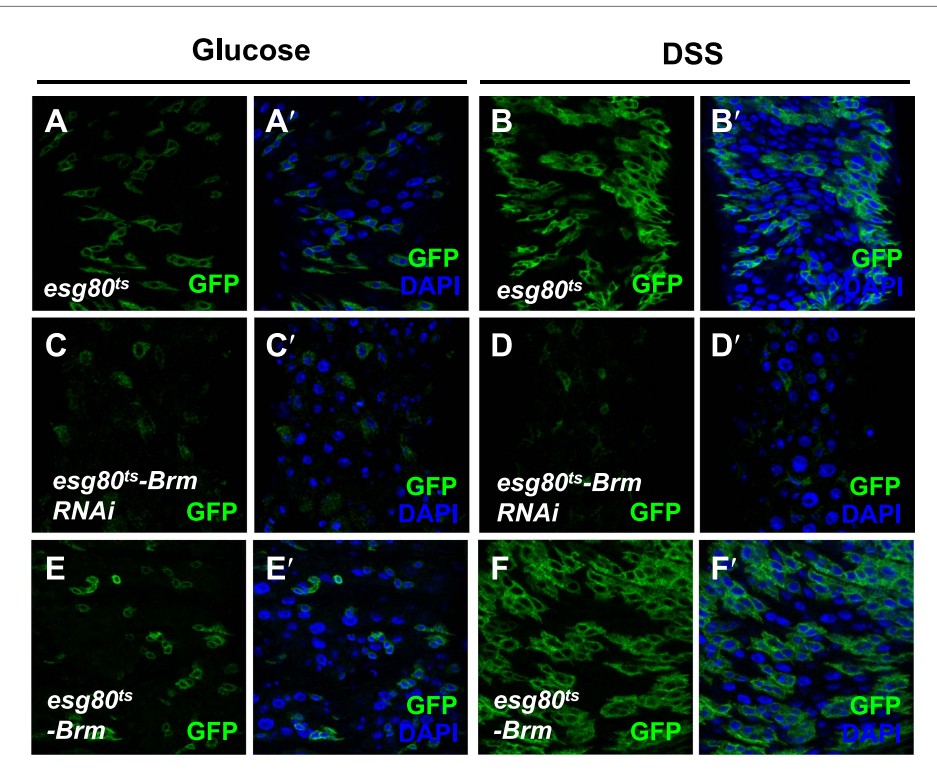

**Figure 3**. Brm was required for midgut regeneration. (**A–F'**) Adult flies expressing *esg80^{ts}-Gal4/UAS-GFP* (*esg80^{ts}*) (**A–B'**), Brm RNAi (*esg80^{ts}-Brm RNAi*) (**C–D'**) or *esg80^{ts}-Gal4/UAS-GFP-Brm* (*esg80^{ts}-Brm*) (**E–F'**) were treated with glucose or DSS. Glucose solution with 3% DSS was fed to the flies (**B–B'**, **D–D'**, and **F–F'**) for 3 days before guts dissection.

**Table 1.** Mass spectrum analysis results

| Protein description | Molecular function | Pep count | Unique Pep count |
|---|---|---|---|
| Yki mass spectrum | | | |
| Brahma (Brm) | ATP-dependent helicase | 11 | 5 |
| Osa | DNA binding | 5 | 4 |
| Sd mass spectrum | | | |
| Brahma associated protein 55kD (Bap55) | Structural constituent of cytoskeleton | 8 | 5 |
| Brahma associated protein 60kD (Bap60) | Protein binding | 5 | 3 |
| Brahma (Brm) | ATP-dependent helicase | 4 | 4 |
| Brahma associated protein 155 kDa (Mor) | Protein binding | 1 | 1 |

To determine whether there are physical interactions between Yki/Sd transcriptional complex and Brm complex and gain further understanding of the regulation mechanism of Brm in regulating ISC proliferation, we immunoprecipitated endogenous Sd or Yki protein in S2 cells using generated rabbit anti-Sd or anti-Yki antibodies, respectively, followed by mass spectrometry (MS) analysis. The corresponding proteins of Brm complex identified in association with Yki (Yki mass spectrum) or Sd (Sd mass spectrum) are listed with the number of peptides identified by mass spectrometry.

cell numbers (compared *Figure 5C,C'',E,E''* with *Figure 5A,A'',K*), suggesting an enhancement of ISC proliferation. Interestingly, Yki overexpression resulted in pronounced hyperplasia of intestine with a thicker intestinal epithelium composed of a multi-layer of tightly packed cells (*Staley and Irvine, 2010*) (compare *Figure 5H* with *Figure 5G*), whereas SdGA expression did not induce such a phenomenon (compare *Figure 5J* with *Figure 5G*), suggesting that Yki and Sd may have distinct mechanisms in regulating ISC proliferation. When Brm was knocked down, ISC proliferation was greatly suppressed (*Figure 5K*, and compare *Figure 5B,B'',D,D'',F,F''* with *Figure 5A,A'',C,C'',E,E''*) with a decreased Dl signal intensity (*Figure 5B',D'*), and the formation of thicker intestinal epithelium induced by Yki overexpression was inhibited (*Figure 5I*). Moreover, similar results were obtained by MARCM analysis of $brm^2$. Overexpression of Yki in control MARCM clones resulted in a significant increase in the cell numbers and in the formation of large clones (*Figure 5L,L',N*), whereas this Yki-induced proliferation was completely blocked in the $brm^2$ clones (*Figure 5M,M',N*). Taken these results together, the depletion of Brm compromised Yki or SdGA overexpression induced ISC proliferation, indicating that Brm functions downstream of Yki–Sd to maintain ISC proliferative ability.

Interestingly, when Yki was expressed in ECs using *Myo1A-Gal4* to induce non-autonomous ISC proliferation, the number of ISCs/EBs was increased with high levels of Brm in the nucleus (compare *Figure 5P–P'''* with *Figure 5O–O'''*), suggesting that Yki-induced non-autonomous ISC proliferation also induces high levels of Brm in nuclei of ISCs and EBs.

## The Hpo signaling regulates Brm protein cleavage

Given that Brm physically interacts with Yki–Sd complex and that the function of Yki–Sd in ISC proliferation requires Brm activity, we sought to determine the underlying mechanism by which the Hpo pathway and Brm regulate ISC proliferation. Interestingly, cotransfection of Brm and Hpo in S2 cells resulted in a lower Brm protein level, suggesting that Brm was destabilized in the presence of Hpo (*Figure 6—figure supplement 1A*). This result raised the concern that the Hpo pathway might regulate Brm activity by controlling its protein stability. Furthermore, Brm cleavage event in which a small protein band at about 100 kD was detected (*Figure 6A*) in the presence of Hpo upon MG132 treatment. To detect whether this small band represents a cleaved Brm fragment, we generated a Brm construct with a Flag tag at the N-terminus and a V5 tag at the C-terminus (referred to as Flag-Brm-V5). Using this construct, we were able to identify a Flag-tagged N-terminal cleavage product about 100 kD and a V5-tagged C-terminal cleavage product about 130 kD in the presence of Hpo (*Figure 6A*). Considering that the molecular weight of full length Brm is about 230 kD, it is possible that Hpo regulates Brm stability by inducing Brm cleavage at only one site. We also found that this cleavage action depended on Hpo protein in a dose-dependent manner, since increasing the dose of Hpo plasmids resulted in an accumulation of the cleaved Brm product and a decrease in the full length Brm protein (*Figure 6B*). A

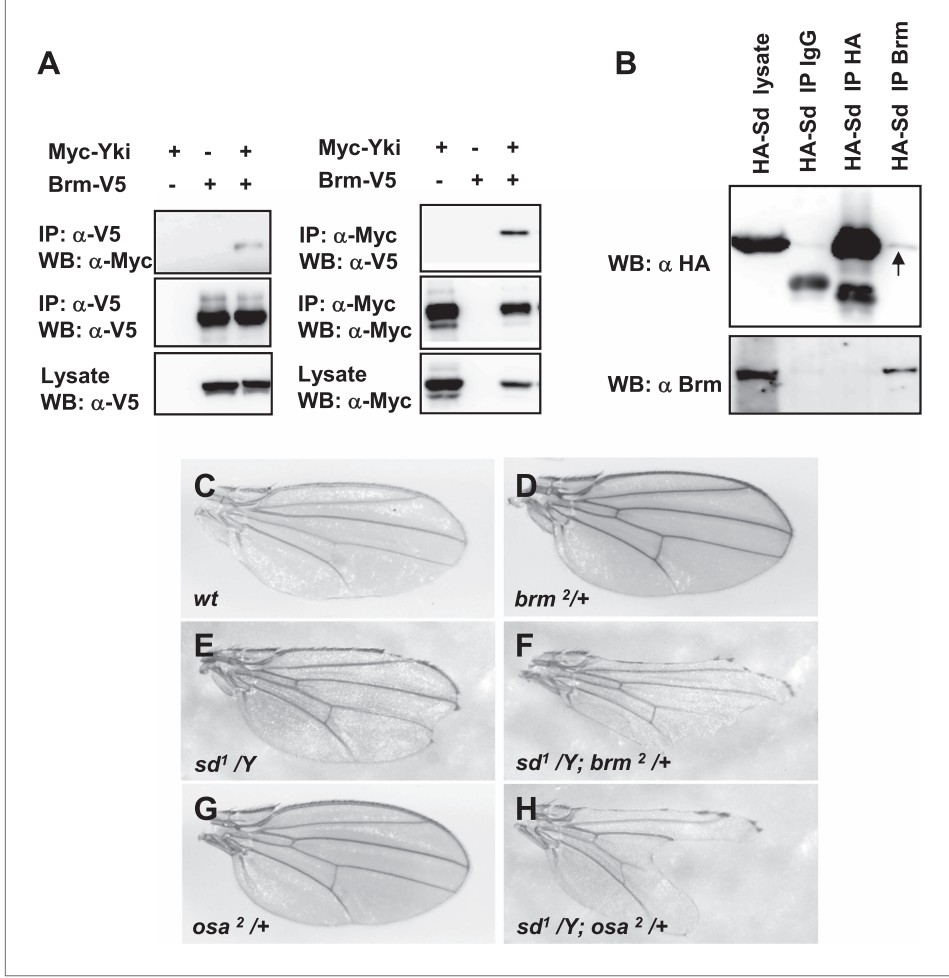

**Figure 4**. Sd and Yki interact with Brm complex components. (**A**) Interaction between overexpressed Myc–Yki and Brm–V5 was detected in S2 cells. Myc–Yki or Brm–V5 was immunoprecipitated with anti-Myc or anti-V5 antibodies. (**B**) Association between HA–Sd and endogenous Brm in vitro. S2 cells were transfected with the HA–Sd. The arrow indicated HA–Sd coimmunoprecipitated with endogenous Brm. (**C–H**) Wild-type male wings (**C**) or hemizygous male wings of null allele *brm²/+* (**D**), or hypomorphic allele *sd¹/Y* (**E**), or double-mutant combinations of *sd¹/Y; brm²/+* (**F**), or hypomorphic allele *osa²/+* (**G**), or combinations of *sd¹/Y; osa²/+* (**H**). See also *Figure 4—figure supplement 1*.

The following figure supplements are available for figure 4:

**Figure supplement 1**. Brm complex associates with Sd.

truncation of Hpo without kinase activity (Hpo-C) did not induce such cleavage (*Figure 6—figure supplement 1B*, lanes 1 and 2), indicating that Hpo kinase domain but not C-terminal regulatory domain induces Brm cleavage. In addition, Hpo-induced Brm cleavage was blocked in the presence of Yki–Sd (*Figure 6C*), suggesting that it was regulated by downstream events of the Hpo signaling pathway.

## Brm cleavage is mediated by Hpo-induced caspase activation

On the basis of above observations, it is feasible to consider that Hpo activates downstream caspases to cleave Brm. In fact, previous studies have implicated a role of the Hpo signaling pathway in caspase activation and cell apoptosis (*Verghese et al., 2012*). This function of Hpo was verified by detecting active caspase 3 expression in wing discs overexpressing Hpo or Hpo-KD (a kinase dead form of Hpo). As shown in *Figure 6—figure supplement 1D–E'''*, the cleaved caspase 3, a functional read-out of initiator caspase activity, was detected only in Hpo overexpressed discs but not in the Hpo-KD overexpressed discs. Taken together, we speculated that Hpo-induced caspase activity might regulate the protein level of Brm.

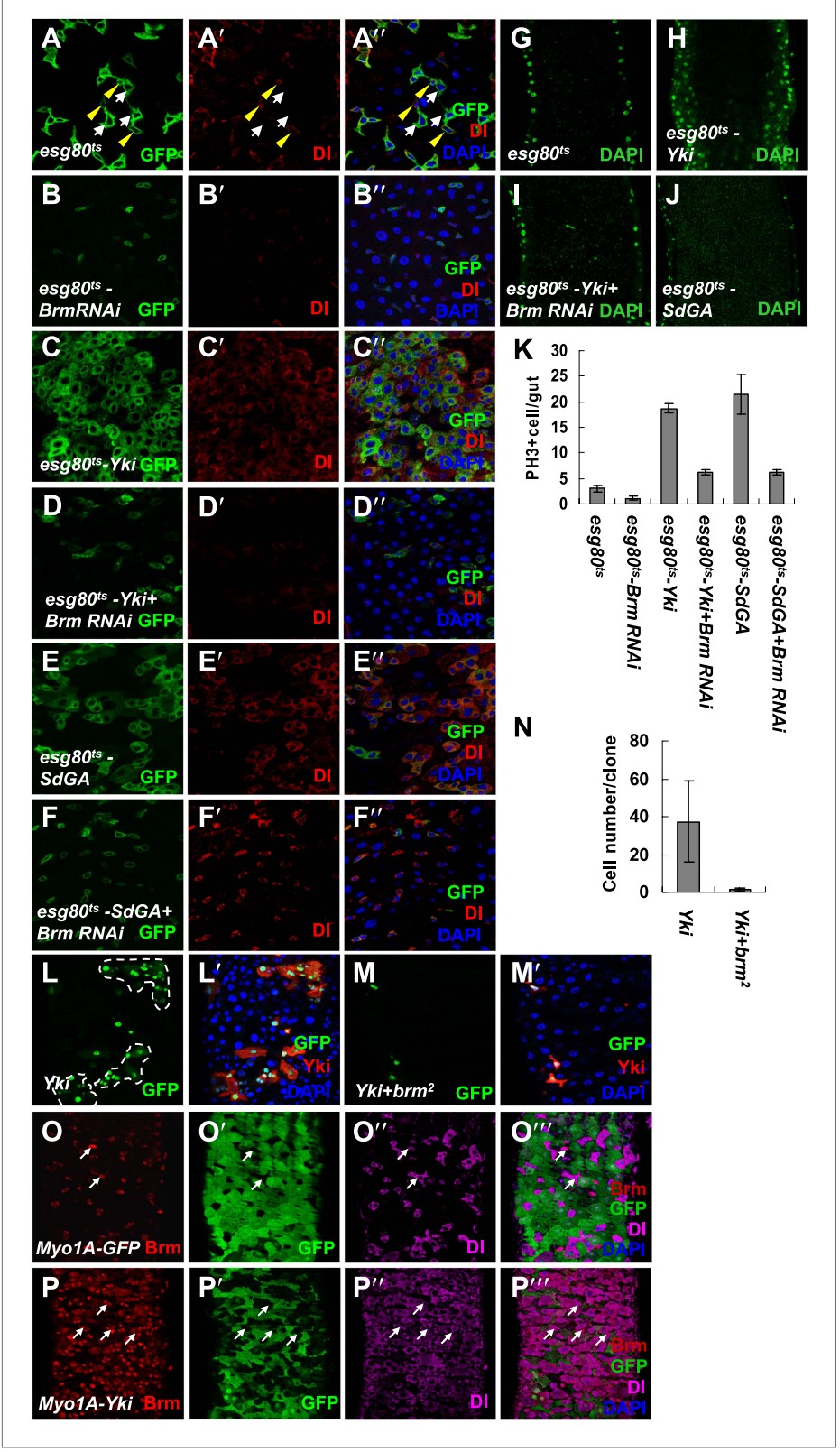

**Figure 5**. Knockdown of Brm blocks Yki/SdGA-induced ISC proliferation. (**A**–**F''**) Adult flies expressing *esg80ts* (**A**–**A''**), *esg80ts-Brm RNAi* (**B**–**B''**), *esg80ts-Yki* (**C**–**C''**), *esg80ts-Yki+Brm RNAi* (**D**–**D''**), *esg80ts-SdGA* (**E**–**E''**), *esg80ts-SdGA +Brm RNAi* (**F**–**F''**) were cultured at 29°C for 8–9 days. Midguts were dissected and immunostained for Dl

*Figure 5. Continued on next page*

*Figure 5. Continued*

(red) and DAPI (blue). White arrows indicated the EBs, and yellow arrowheads indicated the ISCs. (**G**–**J**) Images show an optical cross-section through the center of the intestine, DAPI (green). (**K**) Quantification of PH3 positive mitotic cells of the indicated guts. The results represent the mean ± SEM, n = 10 for each genotype. (**L**–**M′**) Adult midguts containing nuclear localized GFP-labeled control non-tagged form of Yki overexpressed clones (**L** and **L′**) or Yki plus *brm²* clones (**M** and **M′**) were immunostained for Yki (red) and DAPI (blue). Guts were dissected from the adult flies 72 hr after clone induction. (**N**) Quantification of the cell number of *Yki* or *Yki+brm²* clones. 10 guts were counted for each genotype. (**O**–**P‴**) Adult guts of *Myo1A-Gal4 UAS-GFP;tubGal80^ts* control (**O**–**O‴**) or expressing *Myo1A-Gal4 UAS-GFP;tubGal80^ts-Yki* (**P**–**P‴**) were immunostained for Brm (red), Dl (purple), and DAPI (blue). Arrows indicated ISCs with a high endogenous Brm protein level.

To verify this idea, we used the unspecific caspase inhibitor, Z-VAD-FMK, to test whether the activities of caspases are required for Hpo-induced Brm cleavage. We found that the 100 kD cleaved Brm product disappeared on the addition of Z-VAD-FMK (*Figure 6C*), suggesting that the inhibition of caspase activities blocks Brm cleavage. Furthermore, it is known that the *Drosophila* inhibitor of apoptosis protein, Diap1, which is a transcription product of the Hpo pathway target genes (*Zhang et al., 2008*), inhibits caspase activity. Diap1 was cotransfected with Brm and Hpo in S2 cells to inhibit caspase activity. Interestingly, we found that Diap1 cotransfection inhibited Hpo-induced Brm cleavage (*Figure 6D*), indicating that the Hpo regulates Brm cleavage by inducing caspase activity.

To further study the function of caspases during Brm cleavage in details, Hpo and Brm were cotransfected in S2 cells in the presence of inhibitors of mammalian caspase 3, 8, 9, 10, respectively (no commercial *Drosophila* caspase inhibitors were available). As shown in *Figure 6E*, the addition of inhibitor of caspase 3 or caspase 10 completely abolished Hpo-induced Brm cleavage, whereas the addition of other caspase inhibitors only partially affected the cleavage reaction as revealed by the presence of the 100 kD Brm protein fragment. Caspase 10 is an initiator in the extrinsic death-receptor-mediated cell death (*Wachmann et al., 2010*), and caspase 3 is the effector caspase generally believed to carry out the cleavage of nuclear protein substrates. These results suggest that *Drosophila* homologs of caspase 3 and caspase 10 play important roles in Hpo-induced Brm cleavage.

## D718 site is required for Hpo-induced, caspase-dependent, Brm cleavage

In an attempt to identify the cleavage site of Brm, two Brm deletion forms, D1 (Δ601–800 aa) and D2 (Δ694–768 aa), were generated based on previous observations of N- and C-terminal cleavage products (*Figure 6G*). No cleavage reaction was detected for these two Brm deletion forms (*Figure 6—figure supplement 1B*). Mapping of the D2 form using two other deletion forms of Brm, D3 (Δ694–711 aa) and D4 (Δ712–729 aa), indicated that D3 was cleaved but not D4 (*Figure 6—figure supplement 1C*), suggesting that the cleavage site locates within the region of amino acid 712–729. Although no canonical caspase 3 tetra-peptide cleavage site DEVD was found in this region, several aspartic acids that potentially serve as the caspase cleavage sites were identified. To validate these sites, individual aspartic acids were mutated to alanine separately. Interestingly, Brm mutant carrying aspartic acids to alanine mutation at D718 site (Brm^D718A) does not undergo cleavage (*Figure 6F*). In conclusion, Brm protein stability was regulated by Hpo-induced caspase-dependent cleavage at the D718 site.

## The cleavage resistant Brm mutant Brm^D718A promotes ISC proliferation

Given the finding that Brm^D718A was a cleavage resistant Brm mutant (*Figure 6F*), we wondered whether Brm^D718A is an active form of Brm. To test the function of Brm^D718A, we expressed Brm^D718A under the control of *esg80^ts* in ISCs/EBs. An upregulation of ISC/EB (GFP⁺) and PH3⁺ cell numbers was detected in guts expressing Brm^D718A mutant (compare *Figure 7B–B′* with *Figure 7A–A′,J*), whereas expressing wild-type Brm induced a mild increase in the ISC/EB numbers and PH3⁺ cell numbers (*Figure 7G,G′,J*). On coexpression of Brm^D718A and Yki, the number of PH3⁺ cells was further increased, suggesting that ISC proliferation is promoted (*Figure 7C–D′,J*). Furthermore, analysis of 5-bromodeosyuridine (BrdU) incorporation in midguts showed that Brm^D718A overexpression greatly enhanced BrdU ectopic expression, whereas Brm RNAi resulted in a lower proliferative activity (*Figure 7—figure supplement 1A–D′*). Altogether, these results indicate that Brm^D718A promotes ISC proliferation.

To further investigate the function of Brm in ISC proliferation, we overexpressed the truncated form of Brm-N (1–717 aa) or Brm-C (718–1639 aa) in ISCs/EBs under the control of *esg80^ts*. Compared with

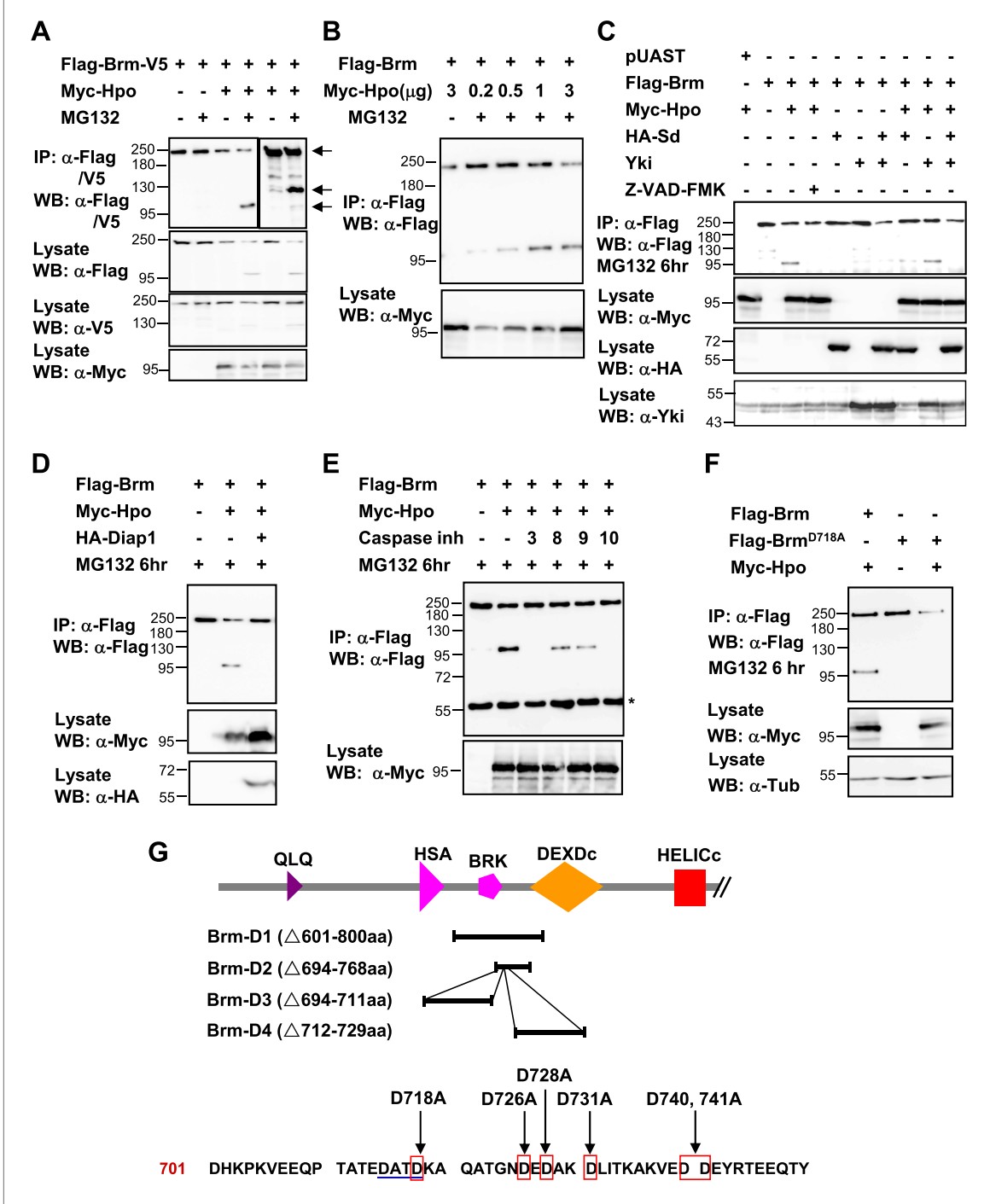

**Figure 6**. Brm is cleaved at the D718 site by Hpo-induced caspase. (**A**) Flag-Brm-V5 was transfected with or without Myc-Hpo. Western blots (anti-Flag or anti-V5) of IP samples were performed to detect the N- or C-terminus of Brm. MG132 was treated 6 hr before harvesting the cells. Arrows indicated the full length Brm (top) and the N- (bottom), C- (middle) terminal cleaved product of Brm. (**B**) 3 µg of Flag-Brm was cotransfected with different dosages of Hpo plasmids in S2 cells, MG132 was treated 6 hr before harvesting the cells. (**C**) Cotransfected Flag-Brm and Myc-Hpo together with Sd/Yki or in the presence of caspase inhibitor Z-VAD-FMK, the cleaved Brm fragments were unable to be detected. Z-VAD-FMK was added to a final concentration of 10 mM for 6 hr. (**D**) S2 cells were transfected with Myc-Hpo and Flag-Brm with HA-Diap1. (**E**) Flag-Brm and Myc-Hpo were cotransfected in S2 cells to induce the cleavage of Brm. Inhibitors of Caspase 3, 8, 9, 10 were added to block the cleavage in a final concentration of 10 mM for 6 hr. Asterisk indicates IgG bands (loading control). (**F**) Brm[D718A] mutation blocked Hpo-induced Brm cleavage in S2 cells. (**G**) A schematic representation of Brm deletions and mutations. Brm-D1 to D4 were the deletions that were used to map the cleavages site of Brm. Brm-D718A/D726A/D728A/D731A/D740,741A were the
*Figure 6. Continued on next page*

*Figure 6. Continued*

mutants generated for mapping the cleavage sites. The novel caspase recognition motif (DATD) in Brm is indicated by a single blue underline including D718 residue. See also *Figure 6—figure supplement 1*.

The following figure supplements are available for figure 6:

**Figure supplement 1**. Brm is cleaved by Hpo-induced caspase.

the wild-type Brm or Brm$^{D718A}$, Brm-C exhibited a weak influence on ISC proliferation while Brm-N did not show any obvious effect (*Figure 7—figure supplement 2A–E,K*). On coexpression with Yki, both Brm-N and Brm-C further promoted Yki-induced ISC proliferation, but not as dramatic as Brm or Brm$^{D718A}$ (*Figure 7—figure supplement 2F–J,K*). To better understand the impact of Brm cleavage on ISC proliferation, rescue experiments were carried out using MARCM approach. We overexpressed Brm, Brm$^{D718A}$, Brm-N and Brm-C in *brm²* MARCM clones and found that all of them were able to partially rescue the growth defect of *brm²* clones to different degrees. Brm$^{D718A}$ possessed the strongest rescue ability, as it generated bigger clones that contain more cells (*Figure 7—figure supplement 2P,L*), while Brm-N and Brm-C only showed weak rescue phenotypes (*Figure 7—figure supplement 2Q,R,L*). These results indicate that Brm cleavage is important for controlling the stability and activity of Brm during ISC proliferation.

## Brm$^{D718A}$ rescues Hpo-restricted ISC proliferation more efficiently than Brm

According to the in vivo observations above, Brm$^{D718A}$ promotes ISC proliferation and exhibits higher activity than wild-type Brm. We speculated that higher activity of Brm$^{D718A}$ might be due to the resistance of Brm$^{D718A}$ against Hpo signaling regulated cleavage. To verify this hypothesis, we coexpressed Hpo and Brm$^{D718A}$ under the control of *esg80$^{ts}$* and found that Brm$^{D718A}$ completely rescued the impairment of ISC proliferation induced by Hpo overexpression. In comparison with the decrease of ISC/EB numbers induced by Hpo overexpression, coexpression of Hpo and Brm$^{D718A}$ exhibited a dramatic increase of ISCs/EBs as well as PH3$^+$ cells in midguts (compare *Figure 7F,F'* with *Figure 7E,E',J*). On the other hand, coexpression of wild-type Brm and Hpo only slightly rescued Hpo-induced decrease of ISCs/EBs (compare *Figure 7H,H'* with *Figure 7E,E'*). In addition, PH3$^+$ cell number was increased when Brm and Hpo were coexpressed (*Figure 7J*), a phenomenon that might be due to an unidentified feedback mechanism of homeostasis in response to Hpo-induced impairment.

To determine whether loss of Hpo expression regulates Brm protein level in midguts, we generated *hpo* null allele *BF33(16)* (*Jin et al., 2012*) MARCM clones in midguts. ECs within the clone regions obtained higher Brm protein levels than ECs outside the clones (*Figure 7I–I'''*), suggesting that Hpo also restricts Brm protein level in ECs. We next expressed Brm or Brm RNAi in *BF33(16)* MARCM clones (*Figure 7—figure supplement 1E–F'*). The growth of the *hpo* null allele clones was not affected by Brm overexpression. Yet, it was suppressed by the knockdown of Brm using RNAi (*Figure 7—figure supplement 1E–F'*), indicating that Brm is required in the loss-of-Hpo-induced intestinal proliferation. Taken together, these results indicate that Brm protein level is restricted by Hpo activity. Brm$^{D718A}$, as an uncleavable form of Brm, bypasses the Hpo restriction to retain its ability to promote ISC proliferation.

To further understand how Brm$^{D718A}$ functions to promote proliferation in other tissues, we investigated the genetic relationship of Brm and Yki in adult eyes under the control of *GMR-Gal4* driver. Overexpression of *UAS-Yki* posterior to the morphogenetic furrow using the *GMR-Gal4* (referred to as *GMR-Yki*) resulted in dramatic eye overgrowth (compare *Figure 7—figure supplement 3D,D'* with *Figure 7—figure supplement 3A,A'*). Consistent with the findings in midguts, expressing wild-type Brm did not significantly affect Yki overexpression induced eye overgrowth (*Figure 7—figure supplement 3E,E'*), yet expressing Brm$^{D718A}$ clearly enhanced *GMR-Yki* induced overgrowth (*Figure 7—figure supplement 3F,F'*). In addition, overexpression of Brm$^{D718A}$ using *hhGal4* caused an upregulation of Diap1 protein levels in the posterior region of the wing discs (compare *Figure 7—figure supplement 4C–C''* with *Figure 7—figure supplement 4A–A''*). Conversely, overexpression of the dominant-negative form of Brm, Brm$^{K804R}$, resulted in a reduction in Diap1 and Bantam levels (*Figure 7—figure supplement 4D–D''* and *Figure 7—figure supplement 4E–F'*). These assays indicate that activated Brm promoted the expression of the Hpo pathway target genes, such as *diap1*.

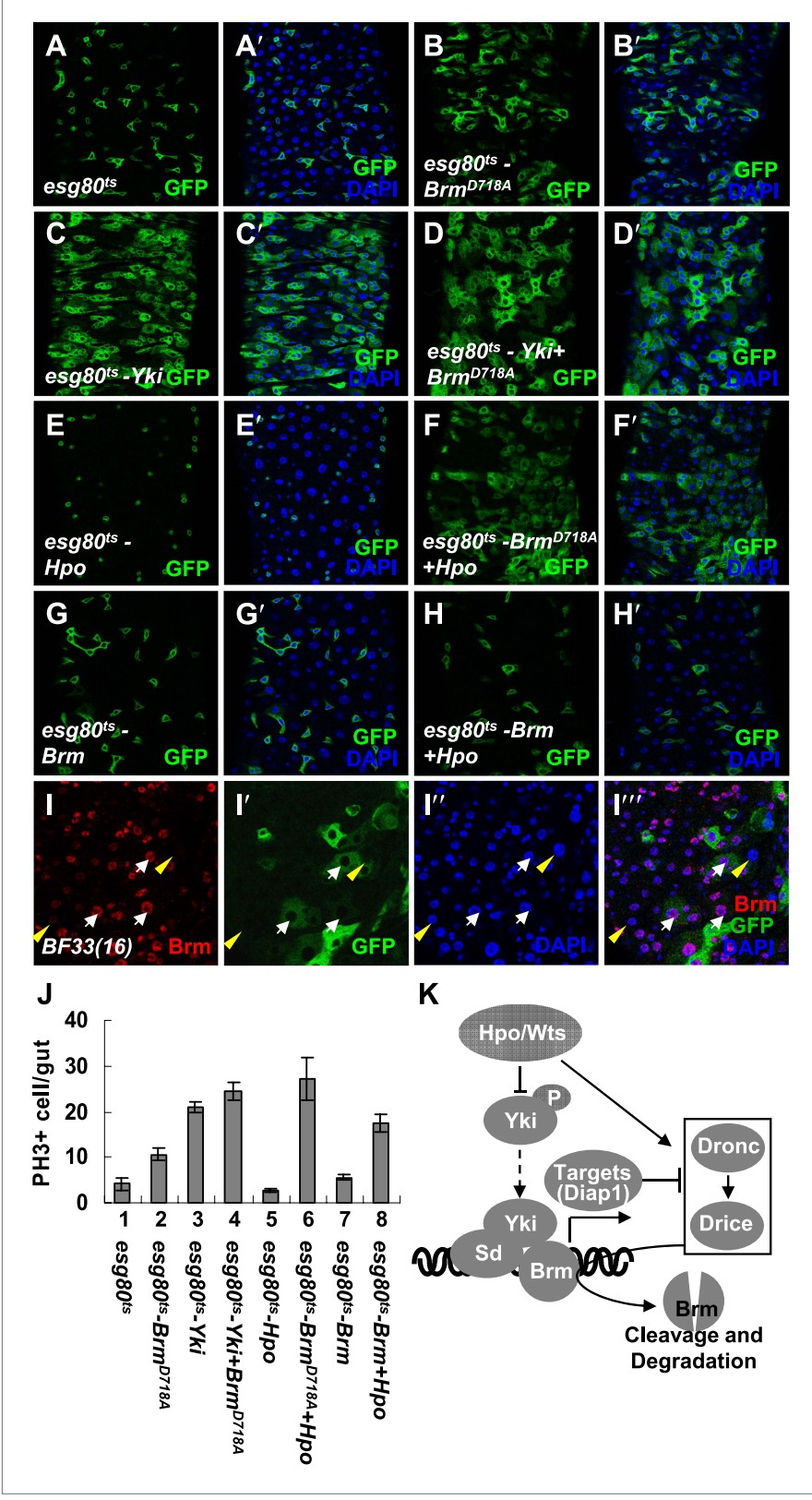

**Figure 7**. The cleavage resistant mutant Brm^D718A promotes ISC proliferation with antagonistic ability against Hpo activity. (**A–H′**) Adult guts of *esg80^ts* control (**A** and **A′**), *esg80^ts-Brm^D718A* (**B** and **B′**), *esg80^ts-Yki* (**C** and **C′**), *esg80^ts-Yki+Brm^D718A* (**D** and **D′**), *esg80^ts-Hpo* (**E** and **E′**), *esg80^ts- Brm^D718A +Hpo* (**F** and **F′**), *esg80^ts-Brm* (**G** and **G′**)
*Figure 7. Continued on next page*

*Figure 7. Continued*

and *esg80^{ts}-Brm+Hpo* (**H** and **H'**) were immunostained for DAPI (blue). (**I–I'''**) Adult midguts containing GFP-labeled MARCM clones of *hpo* null allele (*BF33(16)*). White arrows indicate the ECs in the *BF33(16)* clones, and yellow arrowheads indicate the ECs outside the clones. (**J**) Quantification of PH3 positive mitotic cells of the indicated guts. The results represent the mean ± SEM, n > 10 for each genotype. (**K**) A model of the regulation of Brm protein stability by the Hpo pathway. The Hpo pathway restricts Brm protein level by inducing the activation of caspase to cleave Brm and/or by inhibiting the expression of Yki–Sd target genes, especially *diap1* that inhibits the caspase activity. See also *Figure 7—figure supplements 1–4*.

The following figure supplements are available for figure 7:

**Figure supplement 1**. The cleavage resistance mutant Brm^{D718A} promotes ISCs proliferation.

**Figure supplement 2**. The cleavage products of Brm have low activity in promoting ISC proliferation.

**Figure supplement 3**. Genetic interaction assays between Brm and Yki/Sd in Drosophila eyes.

**Figure supplement 4**. Brm regulates Hpo pathway target genes in wing discs.

## Discussion

SWI/SNF complex subunits regulate the chromatin structure by shutting off or turning on the gene expression during differentiation (*Roberts and Orkin, 2004*). Recently, the findings from several research reports based on the stem cell system reveal important roles of chromatin remodeling complex in stem cell state maintenance (*Lessard et al., 2007*; *Ho et al., 2009*). Our study suggested that the chromatin remodeling activity of Brm complex was required for the proliferation and differentiation of *Drosophila* ISCs. Based on our findings, we propose that Brm is critical for maintaining *Drosophila* intestinal homeostasis (*Figure 2J*). High levels of Brm in the ISC nucleus represent high proliferative ability and are essential for EC differentiation; low levels of Brm in the EC nucleus may be a response for homeostasis. Changes in Brm protein levels resulted in the disruption of differentiation and deregulation of cell proliferation. In line with previous findings in human, the cell-type-specific expression of *Drosophila* homologs BRG1 and BRM were also detected in adult tissues (*Reisman et al., 2005*). BRG1 is mainly expressed in cell types that constantly undergo proliferation or self-renewal, whereas BRM is expressed in other cell types (*Marenda et al., 2004*; *Reisman et al., 2005*). These observations indicate that Brm may act similarly as BRG1 and BRM in controlling proliferation and differentiation.

It is known that the Hpo pathway restricts cell proliferation and promotes cell death at least in two ways: inhibiting the transcriptional co-activator Yki (*Huang et al., 2005*; *Zhang et al., 2008*), and inducing activation of pro-apoptotic genes such as caspases directly (*Verghese et al., 2012*). In our study, we identified a novel regulatory mechanism of the Hpo pathway in maintaining intestinal homeostasis. In this scenario, Brm activity is regulated by the Hpo pathway. In normal physiological conditions, under the control of Hpo signaling, the function of Yki–Sd to promote ISC proliferation is restricted and the pro-proliferation of target genes such as *diap1* that inhibits Hpo-induced caspase activity cannot be further activated (*Figure 7K*). Therefore, Hpo signaling normally functions to restrict cell numbers in the midgut by keeping ISC proliferation at low levels. Yki is enriched in ISCs, but predominantly inactivated in cytoplasm by the Hpo pathway (*Karpowicz et al., 2010*; *Staley and Irvine, 2010*). The knockdown of Yki in ISCs did not cause any phenotype in the midgut (*Karpowicz et al., 2010*), suggesting that Yki is inactivated in ISCs under normal homeostasis. During an injury, Hpo signaling is suppressed or disrupted, Yki translocates into the nuclei to form a complex with Sd (*Karpowicz et al., 2010*; *Ren et al., 2010*; *Shaw et al., 2010*), which may allow Yki–Sd to interact with Brm complex in the nucleus to activate transcriptional targets. Of note, the loss-of-function of Brm resulted in growth defect of ISCs, suggesting that Brm is required for ISC homeostasis and possessing a different role of Brm from Yki in the regulation of ISCs. It is possible that the function of Brm on ISC homeostasis is regulated via other signaling pathways by recruiting other factors. Therefore, different phenotypes induced by the loss-of-function of Brm and Yki in midgut might be due to different regulatory mechanisms. Despite its unique function cooperating with Yki in midgut, that Brm complex is essential for Yki-mediated transcription might be a general requirement for cell proliferation. While this manuscript

was under preparation, Irvine lab reported a genome-wide association of Yki with chromatin and chromatin-remodeling complexes (*Oh et al., 2013*). These results support our model.

Our results also suggest that the interaction between Brm and Yki–Sd transcriptional complex is under tight regulation. The loss of Hpo signaling stabilizes Brm protein, whereas the active Hpo pathway restricts Brm levels by activating *Drosophila* caspases to cleave Brm at the D718 site and inhibiting downstream target gene *diap1* transcription simultaneously. In addition, overexpression of Brm complex components induces only a mild enhancement on midgut proliferation (*Figure 1D,D′* and *Figure 1—figure supplement 2E,E′,G,G′,I,I′*). One possibility is that overexpressing only one of the Brm complex components does not provide full activation of the whole complex; the other possibility is that due to the restriction of the Hpo signaling, as overexpressing Brm$^{D718A}$ mutant protein in ISCs/EBs exhibits a stronger phenotype than expressing the wild-type Brm (*Figure 7B,G*) and coexpression of Brm$^{D718A}$ completely rescues the impairment of Hpo-induced ISC proliferation (*Figure 7F*). D718A mutation blocks the caspase-dependent Brm cleavage and exhibits high activity in promoting ISC proliferation. We have defined a previously unknown, yet essential epigenetic mechanism underlying the role of the Hpo pathway in regulating Brm activity.

It is a novel finding that Brm protein level is regulated by the caspase-dependent cleavage. To focus on the function of Brm cleavage in the presence of cell death signals, we tried to examine the activities of the cleaved Brm fragments. Although in vivo experiments did not show strong activity of Brm N- and C-cleavage products in promoting proliferation of ISCs, the C-terminal fragment of Brm that contains the ATPase domain exhibits a relative higher activity than the N-terminal fragment in ISCs (*Figure 7—figure supplement 2D,E,K*). The cleavage might induce faster degradation of Brm N- and C-terminus, since it was difficult to detect N- or C-fragments of Brm by Western blot analysis without MG132 treatment. It reveals that the degradation events of Brm including both ubiquitination and cleavage at D718 site can be important for Brm functional regulation under different conditions. To this end, the intrinsic signaling(s) may balance the activity of Brm complex through degradation of some important components, such as Brm, to maintain tissue homeostasis. Of note, the cleavage of Brm at D718 is occurred at a novel DATD sequence that is not conserved in human Brm. It has been reported that Cathepsin G, not caspase, cut hBrm during apoptosis (*Biggs et al., 2001*), suggesting that the cleavage regulatory mechanism of Brm is relatively conserved between *Drosophila* and mammals.

In this study, we provide evidence that the Brm complex plays an important role in *Drosophila* ISC proliferation and differentiation and is regulated by multi-levels of Hpo signaling. Our findings indicate that Hpo signaling not only exhibits regulatory roles in organ size control during development but also directly regulates epigenetics through a control of the protein level of epigenetic regulatory component Brm. In mammals, it is known that Hpo signaling and SWI/SNF complex-mediated chromatin remodeling processes play critical roles in tissue development. Malfunction of the Hpo signaling pathway and aberrant expressions of SWI/SNF chromatin-remodeling proteins BRM and BRG1 have been documented in a wide variety of human cancers including colorectal carcinoma (*Reisman et al., 2009*; *Pan, 2010*; *Watanabe et al., 2011*). Thus, our study that implicated a functional link between Hpo signaling pathway and SWI/SNF activity may provide new strategies to develop biomarkers or therapeutic targets.

## Materials and methods

### *Drosophila* stocks and genetics

The following fly stocks were used: *UAS-yki* (*Zhang et al., 2008*), *UAS-HA-Sd* (*Zhang et al., 2008*), *UAS-HA-SdGA* (*Zhang et al., 2008*), *BF33(16)* (*Jin et al., 2012*), *UAS-Flag-Hpo* (*Jin et al., 2012*), *esg-Gal4/UAS-GFP* (*Micchelli and Perrimon, 2006*), *esg-Gal4/UAS-GFP;TubGal80$^{ts}$* (*Micchelli and Perrimon, 2006*), *Myo1A-Gal4/UAS-GFP;TubGal80$^{ts}$* (*Micchelli and Perrimon, 2006*), *w;esgGal4 tubGal80$^{ts}$ UAS-GFP; UAS-flp Act>CD2>Gal4* (*esg$^{ts}$F/O*, a gift from Dr Huaqi Jiang), *esg$^{ts}$Su(H)Z*, *brm$^2$* (Bloomington 3619), *FRT80 brm$^2$*, *sd* hypomorphic allele (*sd$^1$*) (*Zhang et al., 2008*), *osa$^2$* (a gift from Professor Asgar Klebes), *UAS-Brm RNAi* (VDRC 37720, VDRC 37721, and Bloomington 31712), *UAS-Bap60 RNAi* (NIG4303R-1), *UAS-Mor RNAi* (VDRC 6969), *UAS-Osa RNAi* (VDRC 7810), *hhBanGFP*, *UAS-Bap60*, *UAS-Mor*, *UAS-Osa*, *UAS-Brm-N*, *and UAS-Brm-C*. Bap60, Mor, Osa were cloned from the *Drosophila* cDNA. *UAS-Flag-Brm*, *UAS-Flag-Brm$^{K804R}$*, *UAS-Flag-Brm$^{D718A}$*, Brm point mutations and deletions were generated by PCR-based site directed mutagenesis. These cDNA fragments were cloned into the *pUAST* vector. A *pUAST* vector with *attB* sequence inserted upstream of the UAS-binding sites was used to make *pUAST-attB-Brm* and Brm mutants constructs. All plasmids were verified by DNA sequencing. Transgenic flies carrying these constructs were generated.

## MARCM clone analysis

Mutant clones were made using the MARCM system (*Lee and Luo, 2001*). Genotypes for making *brm* mutant clone: *hsflp, tub-Gal4, UAS-GFPnls; tubGal80 FRT80B/FTR80B brm²*. *brm* mutant clone expressing Yki transgene: *hsflp, tub-Gal4, UAS-GFPnls;UAS-Yki/+;tubGal80 FRT80B/FTR80B brm²*. Hpo mutant clone expressing Brm transgene: *yw UAS-GFP hsflp;FRT42D hpo^BF33^/FRT42D tub-Gal80; tubulin-Gal4/ Brm*. Hpo clone expressing Brm RNAi transgene: *yw UAS-GFP hsflp; FRT42D hpo^BF33^/FRT42D tub-Gal80; tubulin-Gal4/UAS-Brm RNAi*. Flies were cultured at 25°C. F1 adult flies with appropriate genotypes were subjected to heat shock at 37°C for 1 hr to induce clone at 5-day-old flies. Then, flies were raised at 25°C for 3 or 8 days before dissection. Clones of more than 10 midguts were scored in each group.

## Temperature-controlled expression

The experiment using *esgGal4 UAS-GFP; tubGal80^ts^* was cultured under 18°C to restrict Gal4 activity. 3-day-old F1 adult flies with appropriate genotypes were then shifted to 29°C for a 7-day incubation to allow inactivation of *Gal80^ts^* and expression of the UAS transgenes or RNAi. 20 female adults with correct genotypes were dissected and subjected to immunostaining. For intestinal stem cell lineage tracing experiment, we used the inducible lineage tracing *esg 80^ts^ F/O* system. 2- to 5-day-old F1 adult flies with correct genotypes cultured at 18°C were then shifted to 29°C to induce the expression of transgenes.

## DSS feeding experiments

Female adult flies (5/6-day-old) were used to perform DSS-treated feeding experiments. Flies were cultured in an empty vial containing a piece of 9 cm² chromatography paper wet with 3% dextran sulfate sodium (MP Biomedicals, Santa Ana, California, United States) in 5% glucose solution for 3 days at 25°C or 29°C.

## Cell culture, transfection and Western blotting

S2 cells were cultured in *Drosophila* Schneider's Medium (Invitrogen, Carlsbad, California, United States) with 10% fetal bovine serum, 100 U/ml of penicillin, and 100 mg/ml of Streptomycin. Plasmid transfection was carried out using LipofectAMINE (Invitrogen) according to manufacturer's instructions. A construct of ubiquitin-Gal4 was cotransfected with *pUAST* expression vectors for all transfection experiments. Immunoprecipitation and Western blot analyses were performed according to standard protocols as previously described (*Jin et al., 2012*). Antibodies used were as follows: mouse anti-Myc (1:5000; Sigma, St. Louis, Missouri, United States), mouse anti-Flag (1:5000; Sigma), mouse anti-V5 (1:5000; Invitrogen), mouse anti-HA (Sigma), rabbit anti-Sd (produced by immunizing rabbits with the peptide of Sd amino acids 208–440), and rabbits anti-Brm (produced by immunizing rabbits with the peptide of Brm amino acids 505–775). The proteasome inhibitor MG132 (Sigma) was solubilized in DMSO and added to a final concentration of 50 μM for 6 hr. Z-VAD-FMK and caspase inhibitors (R&D systems, Minneapolis, Minnesota, United States) were added to a final concentration of 10 μM.

## Immunofluorescence staining

Immunostaining of intestine and S2 cells were carried out as described (*Ren et al., 2010*; *Jin et al., 2012*). Primary antibodies used in this study include mouse anti-Delta (DSHB), mouse anti-Prospero (DSHB, Iowa City, Iowa, United States), rat anti-Ci (1:500), rabbit anti-PH3 (Millipore), rabbit anti-Yki (1:50, produced by immunizing rabbits with the peptide of Yki amino acids 180–418), rabbits anti-Brm (this study), mouse anti-Flag (1:500; Sigma), mouse anti-HA (1:500; Sigma), mouse anti-GFP (1:1000; Santa Cruz, Dallas, Texas, United States), rabbit anti-PDM-1 (1:2000; a gift from Xiaohang Yang, Zhejiang University, Hang Zhou, China).

## Microscopy and data analysis

Fluorescent microscopy was performed on a Leica LAS SP5 confocal microscope; confocal images were obtained using the Leica AF Lite system. Images were processed in Photoshop CS. The GFP⁺ EC cells in *esg80^ts^F/O* gut were counted in one field of view of the posterior midgut near the Malpighian tubules using a 40× objective.

## BrdU incorporation

Adult flies of 7–9 days on food containing BrdU (200 μg/ml in PBS) were mixed into the upper layer and dissected 3 days later. The guts were treated with DNase I for 30 min at 37°C.

## MS samples preparation

Thirty 10-cm dishes of S2 cells were collected and washed twice with cold PBS. The cells were equally divided into two samples and lysed in 2.5 ml lysis buffer (Tris HCL pH 8.0, 50 mM, NaCl 100 mM, NaF 10 mM, $Na_3VO_4$ 1 mM, EDTA 1.5 mM, NP-40 1%, glycerol 10% supplied with protease inhibitor cocktail [Sigma]), and centrifuged at 14,000 rpm for 15 min. Supernatant was transferred to and mixed with 150 μl protein A/G beads (Santa Cruz Biotechnology) at 4°C for 1 hr on a rolling mixer. Then, the mixture was centrifuged at 14,000 rpm for 1 min. Cell supernatant was transferred to a new tube and stored at 4°C.

0.5 mg of Brm antibody was mixed with 0.25 ml of wet beads (use the appropriate antibody/protein A/G combination) in room temperature for 1 hr on a rolling mixer, using serum mixed with beads as control. The beads were washed using 10 vol of Borate Buffer (Sodium Borate pH 9.0, 0.2 mM) twice. 10 μl aliquot of beads was stored on ice (sample 1). The rest of the beads were mixed with solid Dimethyl Pimelimidate Dihydrochloride (final concentration is 20 mM) on the rolling mixer for 30 min at room temperature. Another 10 μl aliquot of beads was collected as sample 2. The rest of the beads were washed twice by equal volume of Ethanolamine after discarding the supernatant. Equal volume of Ethanolamine was added and incubated at room temperature for 2 hr on the rolling mixer, then washed by PBS twice and mixed with the cell supernatant at 4°C for 1 hr on a rolling mixer respectively. The mixture was washed by lysis buffer for three times and stored at −20°C for left experiment.

The coupling of antibody to beads was checked by analyzing the sample 1 and 2 on a SDS gel. After checking the coupling efficiency, for Sd MS, two total samples (serum control and Sd IP sample) were sent for MS directly. For Yki MS, we ran an SDS-PAGE to separate the proteins of Yki IP sample, and then specific bands, which were absent in IgG IP control, were selected for further MS analysis.

## In-gel digestion

For MS fingerprinting, the gel slices were cut out of the preparative Coomassie blue-stained gels, destained with100 mmol/l NH4HCO_3/30% ACN, and then dried completely by centrifugal lyophilization. The dried gel slices were rehydrated with a total of 25 ng of sequencing grade, modified trypsin (Promega, Madison, Wisconsin, United States) in 100 mmol/l ammonium bicarbonate at 4°C for 2 hr. After 20 ml of 50 mmol/l $NH_4HCO_3$, pH 8.3 was supplied, the gel slices were incubated at 37°C for 20 hr. The digest buffer was removed and saved. The gel pieces were then extracted with 200 ml of 60% ACN/0.1%TFA for 15 min with sonication, and the supernatant was removed. The extraction was repeated twice. The three extracts plus the first saved digest buffer were pooled and dried completely by centrifugal lyophilization. This in-gel digestion method was mainly performed according to the method described previously (*Yu et al., 2000*; *Li et al., 2005*) with some modifications as described above.

## MS and protein identification

Peptide mixtures of each gel slice were redissolved in 0.1%TFA, then desalted and concentrated using Stage Tips as reported (*Rappsilber et al., 2007*). Peptide solution was measured using a LTQ Deca XP system (Thermo Finnigan, San Jose, California, United States). HPLC separation was performed with a capillary LC pump. The flow rate of the pump was at 250 μl/min and was about 2 μl/min after split. The mobile phases used for reverse phase were A: 0.1% formic acid in water, pH 3.0, B: 0.1% formic acid in ACN. Peptides were eluted using a 2–35%, 35–90% stepped linear gradient of solvent B in 60 min, 90 min following 90% solvent B in 10 min, and 2% solvent B in 30 min for balance. An ESIIT mass spectrometer (LTQ Deca XP; Thermo Finnigan) was used for peptide detection. The positive ion mode was employed, and the spray voltage was set at 3.4 KV. The spray temperature was set at 200°C for peptides. Collision energy is automatically set by the LTQ Deca XP system. After acquisition of a full scan mass spectrum, 10 MS/MS scans were acquired for the next 10 most intense ions using dynamic exclusion. Peptides and proteins were identified using Turbo Sequest software (Thermo Finnigan), which uses the MS and MS/MS spectra of peptide ions to search against the publicly available Uniport fly database (Version 2011-05-26). The protein identification criteria that we used were based on Delta (CN ≥ 0.1) and Xcorr (one charge ≥ 1.9, two charges ≥ 2.2, three charges ≥ 3.75).

## Acknowledgements

We thank Dr Yingzi Yang, Dr Wei Du, and Dr Dangsheng Li for helpful comments on an earlier version of the manuscript. We also thank Jin Jiang, Dahua Chen, the Bloomington and Vienna Stock Centres, the DSHB, and DGRC for fly stocks.

## Additional information

### Funding

| Funder | Grant reference number | Author |
|---|---|---|
| National Basic Research Program of China (973 Program) | 2010CB912101, 2012CB945001, 2011CB943902 | Lei Zhang, Yun Zhao |
| Strategic Priority Research Program of the Chinese Academy of Sciences | XDA01010406, XDA01010405 | Lei Zhang, Yun Zhao |
| National Natural Science Foundation of China | 31171394, 31371462, 31171414, 31371492 | Lei Zhang, Yun Zhao |
| Hundred Talents Program of the Chinese Academy of Sciences | | Lei Zhang |

The funders had no role in study design, data collection and interpretation, or the decision to submit the work for publication.

### Author contributions

YJ, Conception and design, Acquisition of data, Analysis and interpretation of data, Drafting or revising the article, Contributed unpublished essential data or reagents; JX, Acquisition of data, Analysis and interpretation of data, Drafting or revising the article; M-XY, YL, LH, PL, PZ, ZY, MSH, HJ, YZ, Analysis and interpretation of data, Drafting or revising the article; LZ, Conception and design, Analysis and interpretation of data, Drafting or revising the article

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
