## [Decision Letter]

Thank you for sending your work entitled “Brahma is essential for *Drosophila* intestinal stem cell proliferation and regulated by Hippo signaling” for consideration at eLife. While your article has been favorably peer reviewed by a Senior editor (K VijayRaghavan) and another reviewer, it will be essential for you to respond in full to the comments below as they point to substantive requirements for examining a resubmission.

The Senior editor and the other reviewer discussed their comments before we reached this decision, and the Senior editor has assembled the following comments to help you prepare a revised submission.

Zhang et al. set out to show that “Brahma is essential for *Drosophila* intestinal stem cell proliferation and regulated by Hippo signaling.” There are many features of this paper that make it interesting, suggesting novel mechanisms underlying ISC proliferation. Aspects of this study are thorough with some of the major conclusions well supported by the data. Yet, in other parts, it falls short. A complete resubmission with substantial, robust, and (reasonably rapidly doable) additional experiments could make for an acceptable new submission, which will be examined to see if it retains novelty and whether the hypotheses stand up after examination of results from the experiments suggested.

The principal (potential) novelty and scientific importance of the current study come from the demonstration of the role of Brahma in intestinal stem cell (ISC) regulation and of mechanisms by which its action is regulated. SWI/SNF and Brahma's role has been studied in several other contexts including a variety of stem cell ones. The added value of this study will be if it shows how Brahma mechanistically functions downstream of specific signaling pathways in the ISC context.

The authors start by examining the role of Brm in ISC proliferation. They do this by using an Brm-RNAi under the control of a *escargot-Gal4* driver, conditionally expressed in the adult using a *Gal80*^*ts*^ system (the temperature shift timings should be clearly stated in the results and better detailed in the Materials and methods). The RNAi knockdown is stated in the text as being confirmed by antibody staining, but the relevant figure shows that the RNAi works in the imaginal disc: how do we know that the *escargot-Gal4* is as effective in the adult as the *hh-Gal4* is in the third instar disc? The RNAi phenotype is also averred to be similar to a dominant-negative phenotype: the figures presented do suggest a similarity, though how deep this is not evident from what is presented. The reduction of ISC cells is seen better in the MARCM experiment and this suggests that the RNAi could be effective as assumed. In sum, the conclusion of a role for Brm in ISC proliferation would have been stronger if an independent ISC antibody marker were used to verify the RNAi result. This important point needs to be addressed. As it stands, the principal assertion of the first figure is weak, thereby weakening the rest that follows.

The authors next go on to examine the role of Brm in damage induced midgut regeneration. A couple of points here are puzzling. First, the *Gal80*^*ts*^ does not seem to be used here. Presumably then, the RNAi will be active throughout development. Does this not alter the possibilities for interpreting the data? Figure 1O and Figure 1 do not look similar. Can the authors address this? Could it not be that the ISCs are very few (as shown earlier) and that the absence of proliferation of these few ISC upon DSS induction is due to an earlier role for Brm rather than an additional later role on damage induced midgut regeneration?

Next, the authors examine if Brm is required for EC differentiation. They are led to this by the determination that in *brm*^*2*^ mutant clones they see “that *brm*^*2*^ clones only contained one small nuclear cell (Figure 1G–G').” Brm is expressed at low levels in ECs. The lineage trace method is not well explained anywhere in the text. Presumably they imply that this method allows the removal of even this low-level of Brm in early-ECs (35) and they then examine the effect on EC differentiation. This point needs to be clarified and clearly demonstrated. If, on the other hand (and this seems to be the case), they suggest that the ISC that are present (without Brm) do not proliferate and therefore there are fewer ECs as seen by the absence of PDM-1, then this suggests a role for Brm in ISCs and not necessarily in EC differentiation too. A clean way to examine the role of Brm in EC differentiation will be to down-regulate Brm specifically in early-ECs but not in ISCs. The authors could examine if this is feasible using conditional *Gal4* drivers.

The authors then go on to examine the interaction of Brm with the Yki-Sd complex. This is done in S2 cells using pull-down of endogenously expressed Yki or Sd proteins followed by MS. They also use over-expressed tagged proteins to show co-immunoprecipitation. This section, with respect to the main theme of the paper, raises concern. So far, we have epistasis showing that Brm is downstream of Yki-Sd. Now we are shown physical interaction in another very different context (S2) cells (similar results have also been recently published by the Ken Irvine Lab). These S2 results are also interpreted quantitatively and this interpretation transferred to an in vivo context: it is not clear how one can do this, coming as it does from a rather artificial and experimentally highly-manipulated cell-culture context. If the authors do not show in vivo interaction by high-resolution imaging methods they are not adding anything too much to what we have already from the Irvine lab paper. Just reporting this as an independent conformation of the Irvine lab results is confirmatory, but that can be stated in a less elaborate way. Similarly, the genetic interactions of Brm with Sd are very interesting from a genetic perspective, but other that hinting tantalizingly to a similar interaction in ISCs we do not know if this wing-blade interaction applies in this context. Indeed does the mild phenotype from Sd over-expression not suggest that Sd may not have a major role in ISC differentiation, although Yki does?

Again, the fact that the EGFR gain- and Notch-RNAi phenotypes are not altered by Brm loss of function does not necessarily mean that the EGFR and Notch pathways function independent of Brm. Several more gain and loss of function studies and colocalization (or otherwise) of effectors will be needed before any conclusion is reached. Indeed, by bringing in this section too the authors do not help themselves. Leaving it out will not change the main point they are trying to make.

Finally, the result that the Hippo-kinase activity regulates a caspase, which cleaves Brm and that the Cleavage resistant Brm illuminates a regulatory role for Brm is very interesting. Indeed this is the most valuable part of the paper. Unfortunately, it is not shown as being substantively related to the ISC context. This is a fundamental gap that needs to be filled. The authors need further evidence to address the significance of the Brm cleavage during ISC proliferation. The activity of Brm N and C terminus was mentioned as “data not shown”. It will be important to show them clearly and interpret them. The activity, in ISCs, of both cleaved products of Brm should be analyzed in wild-type, Brm null and Yki over-expression backgrounds. The Hippo dependence of cleavage in ISCs also needs demonstration. As presented, while these last sets of results (on Brm cleavage) are quite interesting from a Brm regulation viewpoint, they are not fundamental to the claims the paper starts out to make as they are not substantively demonstrated in ISCs.

---

## [Author Response]

*The authors start by examining the role of Brm in ISC proliferation. They do this by using an Brm-RNAi under the control of a* escargot-Gal4 *driver, conditionally expressed in the adult using a* Gal80^ts^
*system (The temperature shift timings should be clearly stated in the Results and better detailed in the Materials and methods). The RNAi knockdown is stated in the text as being confirmed by antibody staining, but the relevant figure shows that the RNAi works in the imaginal disc: how do we know that the* escargot*-*GaL4 *is as effective in the adult as the* hh-Gal4 *is in the third instar disc? The RNAi phenotype is also averred to be similar to a dominant-negative phenotype: The figures presented do suggest a similarity, though how deep this is not evident from what is presented. The reduction of ISC cells is seen better in the MARCM experiment and this suggests that the RNAi could be effective as assumed. In sum, the conclusion of a role for Brm in ISC proliferation would have been stronger if an independent ISC antibody marker were used to verify the RNAi result. This important point needs to be addressed. As it stands, the principal assertion of the first figure is weak, thereby weakening the rest that follows*.

We now added a description of the *Gal80*^*ts*^ system in the main text and a detailed experimental protocol in the Materials and methods section. To clarify the point that the Brm RNAi works efficiently in the ISCs/EBs, we stained the endogenous Brm protein in the control wild type *esg80*^*ts*^ gut and *esg80*^*ts*^*-Brm* RNAi guts (Figure 1—figure supplement 1). Compared with the control (Figure 1—figure supplement 1), Brm levels in the ISCs/EBs were efficiently reduced by Brm RNAi (Figure 1—figure supplement 1). Of note, the weak signal of Brm in Figure 1—figure supplement 1 is from the large nuclear ECs. Brm dominant-negative form K804R does not possess its ATPase activity, yet does not affect the complex assembly. We predict that it will have a weaker function than Brm RNAi. To verify the RNAi results, we had used Delta (Dl), which is a unique ISC marker, to mark ISC in *esg80*^*ts*^*-Brm* RNAi experiments (Figure 5). Brm RNAi expression decreased the Dl^+^ cell number in Figure 5’.

*The authors next go on to examine the role of Brm in damage induced midgut regeneration. A couple of points here are puzzling. First, the* Gal80^ts^
*does not seem to be used here. Presumably then, the RNAi will be active throughout development. Does this not alter the possibilities for interpreting the data?*
Figure 1
*and*
Figure 1
*do not look similar. Can the authors address this? Could it not be that the ISCs are very few (as shown earlier) and that the absence of proliferation of these few ISC upon DSS induction is due to an earlier role for Brm rather than an additional later role on damage induced midgut regeneration*?

In fact, we used both *esgGal4* and *esg80*^*ts*^ to examine the role of Brm in damage induced midgut regeneration in the experiment. Both gave similar results. In the previous manuscript, we showed the *esgGal4* data; to avoid misinterpretation of developmental difference, we now replaced the *esgGal4* data with the *esg80*^*ts*^ data (Figure 3). The old Figure 1 and Figure 1 do not look similar due to: 1) the difference in GFP protein distribution for *esgGal4*-GFP (in both nucleus and cytoplasm) and *esg80*^*ts*^-GFP (mainly in the cytoplasm) and 2) different culture conditions and food conditions of these two experiments. The flies used for Figure 1 were fed with normal food, but the flies used for DSS-treated experiments (Figure 3) were fed with glucose and water in order to make the final concentration of DSS steady at 3%. During the experimental operation, we noticed that the guts expressing Brm RNAi from the DSS-treated experiments (Figure 3) were more fragile and much thinner than the guts from the experiments of Figure 1 using the same esg80^ts^ driver. Therefore, based on these results, we believe that the DSS-induced midgut regeneration can be blocked by the loss of Brm, but not due to the earlier role of Brm knockdown in development.

*Next, the authors examine if Brm is required for EC differentiation. They are led to this by the determination that in* brm^2^
*mutant clones they see “that* brm^2^
*clones only contained one small nuclear cell (*Figure 1*).” Brm is expressed at low levels in ECs. The lineage trace method is not well explained anywhere in the text. Presumably they imply that this method allows the removal of even this low-level of Brm in early-ECs (*[35]*) and they then examine the effect on EC differentiation. This point needs to be clarified and clearly demonstrated. If, on the other hand (and this seems to be the case) they suggest that the ISC that are present (without Brm) do not proliferate and therefore there are fewer ECs as seen by the absence of PDM-1, then this suggests a role for Brm in ISCs and not necessarily in EC differentiation too. A clean way to examine the role of Brm in EC differentiation will be to down-regulate Brm specifically in early-ECs but not in ISCs. The authors could examine if this is feasible using conditional* Gal4 *drivers*.

As suggested, we gave a description of the lineage trace method in the revised manuscript. To better understand the role of Brm in EC differentiation, we tried to find probable drivers for lineage tracing of early-ECs, but unfortunately, no suitable early-ECs specific conditional *Gal4* drivers is reported yet (Myo1A driver is also expressed in mature ECs (Jiang and Edgar, 2009)). Instead, we analyzed the function of Brm in ISCs division by detecting the Su(H)lacZ, a marker of EB cells. We found that Brm RNAi did not block EBs formation, and added these data to Figure 1—figure supplement 1 with a description in the revised manuscript. Together with our esgF/O lineage tracing data, we speculated that, if Brm do not affect the differentiation from EBs to ECs, EBs expressing esgF/O-Brm RNAi will finally differentiate into ECs like control group (Figure 2, Figure 2—figure supplement 1). However, we observed very few ECs in the Brm RNAi guts (Figure 2 and Figure 2—figure supplement 1) even though relatively a large number of EB cells was observed (Figure 1—figure supplement 1), indicating that the differentiation from EBs to ECs was affected. The other evidence that supports the conclusion is that Brm overexpression promotes precocious differentiation of ECs at day 2 after heat shock (Figure 2 compared with Figure 2), while the GFP^+^ cells in control group do not form EC at this time point.

*The authors then go on to examine the interaction of Brm with the Yki-Sd complex. This is done in S2 cells using pull-down of endogenously expressed Yki or Sd proteins followed by MS. They also use over-expressed tagged proteins to show co-immunoprecipitation. This section, with respect to the main theme of the paper, raises concern. So far, we have epistasis showing that Brm is downstream of Yki-Sd. Now we are shown physical interaction in another very different context (S2) cells (similar results have also been recently published by the Ken Irvine Lab). These S2 results are also interpreted quantitatively and this interpretation transferred to an in vivo context: it is not clear how one can do this, coming as it does from a rather artificial and experimentally highly-manipulated cell-culture context. If the authors do not show in vivo interaction by high-resolution imaging methods they are not adding anything too much to what we have already from the Irvine lab paper. Just reporting this as an independent conformation of the Irvine lab results is confirmatory, but that can be stated in a less elaborate way. Similarly, the genetic interactions of Brm with Sd are very interesting from a genetic perspective, but other that hinting tantalizingly to a similar interaction in ISCs we do not know if this wing-blade interaction applies in this context. Indeed does the mild phenotype from Sd over-expression not suggest that Sd may not have a major role in ISC differentiation, although Yki does*?

We agree with reviewers’ opinions. As suggested, we weakened this part in the revised manuscript. The function of Brm on cell proliferation may be a general requirement even though the regulation is cell type specific. In tissues other than guts, such as wing discs (34), knockdown of Brm inhibits Yki-induced proliferation, suggesting that Brm is generally required for Yki activity. Knockdown of Yki in ISCs does not lead to any obvious phenotypes even though Yki protein is enriched in ISCs (23); knockdown of Yki in tissues, like wing discs, blocks the growth of the cells (15). This piece of evidence suggests that distinct regulatory mechanisms exist in the ISCs to control Yki’s activity.

The genetic interactions of *brm* with *sd* in wings together with their physical interactions also suggested that Sd activity is generally regulated by Brm, which is novel. In midguts, when we generated *sd* null MARCM clones in guts, we found that, after 14 days of heat shock, the size of *sd* mutant clones were larger than the wild type control clones, suggesting that loss of Sd promotes proliferation of the gut cells in normal condition, a phenomenon not similar to that by Yki knockdown. When we overexpressed Yki in the *sd* MARCM clones, the clone size is much smaller than the Yki overexpressed clones. These data suggested that, Sd is involved in hyper-active Yki induced proliferation of ISCs, but it is also a default transcriptional repressor in normal condition, which is consistent with the findings recently published by DJ Pan’s lab (Koontz et al., 2013) and by our lab (Guo et al., 2013). We observed that SdGA (an active form of Sd) over-expression in ISCs/EBs had a mild phenotype (mono-layer) than Yki over-expression, suggesting that Yki plays a more extensive role in ISC proliferation through cooperation with other factors except Sd.

*Again, the fact that the EGFR gain- and Notch-RNAi phenotypes are not altered by Brm loss of function does not necessarily mean that the EGFR and Notch pathways function independent of Brm. Several more gain and loss of function studies and colocalization (or otherwise) of effectors will be needed before any conclusion is reached. Indeed, by bringing in this section too the authors do not help themselves. Leaving it out will not change the main point they are trying to make*.

We agree with the reviewers’ opinions. To put less emphasis on this point, we have deleted this part.

*Finally, the result that the Hippo-kinase activity regulates a caspase, which cleaves Brm and that the Cleavage resistant Brm illuminates a regulatory role for Brm is very interesting. Indeed this is the most valuable part of the paper. Unfortunately, it is not shown as being substantively related to the ISC context. This is a fundamental gap that needs to be filled. The authors need further evidence to address the significance of the Brm cleavage during ISC proliferation. The activity of Brm N and C terminus was mentioned as “data not shown”. It will be important to show them clearly and interpret them. The activity, in ISCs, of both cleaved products of Brm should be analyzed in wild-type, Brm null and Yki over-expression backgrounds. The Hippo dependence of cleavage in ISCs also needs demonstration. As presented, while these last set of results (on Brm cleavage) are quite interesting from a Brm regulation viewpoint, they are not fundamental to the claims the paper starts out to make as they are not substantively demonstrated in ISCs*.

We thank the reviewers for these suggestions. As suggested, we comprehensively analyzed the activity of the cleaved products of Brm in wild-type, Yki overexpression, and Brm null backgrounds, and added these data to Figure 7—figure supplement 2. In wild-type background, Brm-C induces ISC proliferation by increasing the number of ISCs/EBs and PH3^+^ cells, but not as strong as Brm^D718A^, while Brm-N didn’t exhibit any obvious change (Figure 7—figure supplement 2).

Similar results were also observed in Yki overexpression backgrounds. Compared with Yki overexpression in ISCs/EBs (Figure 7—figure supplement 2), overexpressing wild type Brm or Brm^D718A^ together with Yki further promoted the proliferation of ISCs and increased the PH3^+^ cell number dramatically (Figure 7—figure supplement 2). Brm-N and Brm-C were able to promote the Yki induced ISCs proliferation mildly (Figure 7—figure supplement 2).

We overexpressed Brm, Brm^D718A^, Brm-N and Brm-C in *brm* null MARCM clones. After induction for 3 or 10 days, all of them were able to partially rescue the growth defect of *brm* null clones in different degrees, and formed larger clones containing more cells than the *brm*^*2*^ clones. Brm^D718A^ exhibited the strongest activity as shown by forming bigger clones (Figure 7—figure supplement 2), and Brm cleavage products, Brm-N or C, only had weak rescue ability (Figure 7—figure supplement 2).

We also tried to check the cleavage event of Brm in ISCs and EBs in midgut. We dissected more than 100 guts each to detect the cleavage of endogenous Brm or the overexpressed Flag-Brm/Flag- Brm^D718A^ in ISCs/EBs with or without Hpo co-expression by Western blot. Unfortunately, we failed to detect any Brm signal. It may be due to the fact that Brm protein levels are high in ISCs and EBs, yet the gut only contains a small number of ISCs and EBs, and most of the gut cells are ECs with low Brm protein levels.